# A Gradient Flow Framework For Analyzing Network Pruning

**Ekdeep Singh Lubana & Robert P. Dick**
EECS Department
University of Michigan
Ann Arbor, MI 48105, USA
`{eslubana, dickrp}@umich.edu`

## Abstract

Recent network pruning methods focus on pruning models early-on in training. To estimate the impact of removing a parameter, these methods use importance measures that were originally designed to prune trained models. Despite lacking justification for their use early-on in training, such measures result in surprisingly low accuracy loss. To better explain this behavior, we develop a general framework that uses gradient flow to unify state-of-the-art importance measures through the norm of model parameters. We use this framework to determine the relationship between pruning measures and evolution of model parameters, establishing several results related to pruning models early-on in training: (i) magnitude-based pruning removes parameters that contribute least to reduction in loss, resulting in models that converge faster than magnitude-agnostic methods; (ii) loss-preservation based pruning preserves first-order model evolution dynamics and its use is therefore justified for pruning minimally trained models; and (iii) gradient-norm based pruning affects second-order model evolution dynamics, such that increasing gradient norm via pruning can produce poorly performing models. We validate our claims on several VGG-13, MobileNet-V1, and ResNet-56 models trained on CIFAR-10/CIFAR-100.

## 1 Introduction

The use of Deep Neural Networks (DNNs) in intelligent edge systems has been enabled by extensive research on model compression. "Pruning" techniques are commonly used to remove "unimportant" filters to either preserve or promote specific, desirable model properties. Most pruning methods were originally designed to compress trained models, with the goal of reducing inference costs only. For example, Li et al. (2017); He et al. (2018) proposed to remove filters with small $\ell 1/\ell 2$ norm, thus ensuring minimal change in model output. Molchanov et al. (2017; 2019); Theis et al. (2018) proposed to preserve the loss of a model, generally using Taylor expansions around a filter's parameters to estimate change in loss as a function of its removal.

Recent works focus on pruning models at initialization (Lee et al. (2019; 2020)) or after minimal training (You et al. (2020)), thus enabling reduction in both inference and training costs. To estimate the impact of removing a parameter, these methods use the same importance measures as designed for pruning trained models. Since such measures focus on preserving model outputs or loss, Wang et al. (2020) argue they are not well-motivated for pruning models early-on in training. However, in this paper, we demonstrate that if the relationship between importance measures used for pruning trained models and the evolution of model parameters is established, their use early-on in training can be better justified.

In particular, we employ gradient flow[1] to develop a *general framework that relates state-of-the-art importance measures used in network pruning through the norm of model parameters*. This framework establishes the relationship between regularly used importance measures and the evolution of a model's parameters, thus demonstrating why measures designed to prune *trained* models also

---

[1]gradient flow refers to gradient descent with infinitesimal learning rate; see Equation 6 for a short primer.

perform well *early-on in training*. More generally, our framework enables better understanding of what properties make a parameter dispensable according to a particular importance measure. Our findings follow. (i) Magnitude-based pruning measures remove parameters that contribute least to reduction in loss. This enables magnitude-based pruned models to achieve faster convergence than magnitude-agnostic measures. (ii) Loss-preservation based measures remove parameters with the least tendency to change, thus preserving first-order model evolution dynamics. This shows the use of loss-preservation is justified for pruning models early-on in training as well. (iii) Gradient-norm based pruning is linearly related to second-order model evolution dynamics. Increasing gradient norm via pruning for even slightly trained models can permanently damage earlier layers, producing poorly performing architectures. This behavior is a result of aggressively pruning filters that maximally increase model loss. We validate our claims on several VGG-13, MobileNet-V1, and ResNet-56 models trained on CIFAR-10 and CIFAR-100.

## 2 RELATED WORK

Several pruning frameworks define importance measures to estimate the impact of removing a parameter. Most popular importance measures are based on parameter magnitude (Li et al. (2017); He et al. (2018); Liu et al. (2017)) or loss preservation (Molchanov et al. (2019; 2017); Theis et al. (2018); Gao et al. (2019)). Recent works show that using these measures, models pruned at initialization (Lee et al. (2019); Wang et al. (2020); Hayou et al. (2021)) or after minimal training (You et al. (2020)) achieve final performance similar to the original networks. Since measures for pruning trained models are motivated by output or loss preservation, Wang et al. (2020) argue they may not be well suited for pruning models early-on in training. They thus propose GraSP, a measure that promotes preservation of parameters that increase the gradient norm.

Despite its success, the foundations of network pruning are not well understood. Recent work has shown that good "subnetworks" that achieve similar performance to the original network exist within both trained (Ye et al. (2020)) and untrained models (Frankle & Carbin (2019); Malach et al. (2020); Pensia et al. (2020)). These works thus prove networks *can* be pruned without loss in performance, but do not indicate *how* a network should be pruned, i.e, which importance measures are preferable. In fact, Liu et al. (2019) show reinitializing pruned models before retraining rarely affects their performance, indicating the consequential differences among importance measures are in the properties of architectures they produce. Since different importance measures perform differently (see Appendix E), analyzing popular measures to determine which model properties they tend to preserve can reveal which measures lead to better-performing architectures.

From an implementation standpoint, pruning approaches can be placed in two categories. The first, structured pruning (Li et al. (2017); He et al. (2018); Liu et al. (2017); Molchanov et al. (2019; 2017); Gao et al. (2019)), removes entire filters, thus preserving structural regularity and directly improving execution efficiency on commodity hardware platforms. The second, unstructured pruning (Han et al. (2016b); LeCun et al. (1990); Hassibi & Stork (1993)), is more fine-grained, operating at the level of individual parameters instead of filters. Unstructured pruning has recently been used to reduce computational complexity as well, but requires specially designed hardware (Han et al. (2016a)) or software (Elsen et al. (2020)) that are capable of accelerating sparse operations. By clarifying benefits and pitfalls of popular importance measures, our work aims to ensure practitioners are better able to make informed choices for reducing DNN training/inference expenditure via network pruning. Thus, while results in this paper are applicable in both structured and unstructured settings, our experimental evaluation primarily focuses on structured pruning early-on in training. Results on unstructured pruning are relegated to Appendix H.

## 3 PRELIMINARIES: CLASSES OF STANDARD IMPORTANCE MEASURES

In this section, we review the most successful classes of importance measures for network pruning. These measures will be our focus in subsequent sections. We use bold symbols to denote vectors and italicize scalar variables. Consider a model that is parameterized as $\mathbf{\Theta}(t)$ at time $t$. We denote the gradient of the loss with respect to model parameters at time $t$ as $\mathbf{g}(\mathbf{\Theta}(t))$, the Hessian as $\mathbf{H}(\mathbf{\Theta}(t))$, and the model loss as $L(\mathbf{\Theta}(t))$. A general model parameter is denoted as $\theta(t)$. The importance of a set of parameters $\mathbf{\Theta}_p(t)$ is denoted as $I(\mathbf{\Theta}_p(t))$.

**Magnitude-based measures:** Both $\ell 1$ norm (Li et al. (2017)) and $\ell 2$ norm (He et al. (2018)) have been successfully used as magnitude-focused importance measures and generally perform equally well. Due to its differentiability, $\ell 2$ norm can be analyzed using gradient flow and will be our focus in the following sections.

$$I(\mathbf{\Theta}_p(t)) = \|\mathbf{\Theta}_p(t)\|_2^2 = \sum_{\theta_i \in \mathbf{\Theta}_p} (\theta_i(t))^2. \tag{1}$$

**Loss-preservation based measures:** These measures generally use a first-order Taylor decomposition to determine the impact removing a set of parameters has on model loss. Most recent methods (Molchanov et al. (2019; 2017); Ding et al. (2019); Theis et al. (2018)) for pruning trained models are variants of this method, often using additional heuristics to improve their performance.

$$L\left(\mathbf{\Theta}(t) - \mathbf{\Theta}_p(t)\right) - L\left(\mathbf{\Theta}(t)\right) \approx -\mathbf{\Theta}_p^T(t)\mathbf{g}(\mathbf{\Theta}(t)). \tag{2}$$

The equation above implies that the loss of a pruned model is *higher (lower)* than the original model if parameters with a *negative (positive)* value for $\mathbf{\Theta}_p^T(t)\mathbf{g}(\mathbf{\Theta}(t))$ are removed. Thus, for *preserving* model loss, the following importance score should be used.

$$I(\mathbf{\Theta}_p(t)) = \left|\mathbf{\Theta}_p^T(t)\mathbf{g}(\mathbf{\Theta}(t))\right|. \tag{3}$$

**Increase in gradient-norm based measures:** Wang et al. (2020) argue loss-preservation based methods are not well-motivated for pruning models early-on in training. They thus propose GraSP, an importance measure that prunes parameters whose removal increases the gradient norm and can enable fast convergence for a pruned model.

$$\|\mathbf{g}\left(\mathbf{\Theta}(t) - \mathbf{\Theta}_p(t)\right)\|_2^2 - \|\mathbf{g}\left(\mathbf{\Theta}(t)\right)\|_2^2 \approx -2\mathbf{\Theta}_p^T(t)\mathbf{H}(\mathbf{\Theta}(t))\mathbf{g}(\mathbf{\Theta}(t)). \tag{4}$$

The above equation implies that the gradient norm of a pruned model is *higher* than the original model if parameters with a *negative* value for $\mathbf{\Theta}_p^T(t)\mathbf{H}(\mathbf{\Theta}(t))\mathbf{g}(\mathbf{\Theta}(t))$ are removed. This results in the following importance score.

$$I(\mathbf{\Theta}_p(t)) = \mathbf{\Theta}_p^T(t)\mathbf{H}(\mathbf{\Theta}(t))\mathbf{g}(\mathbf{\Theta}(t)). \tag{5}$$

As mentioned before, these importance measures were introduced for pruning trained models (except for GraSP), but are also used for pruning models early-on in training. In the following sections, we revisit the original goals for these measures, establish their relationship with evolution of model parameters over time, and provide clear justifications for their use early-on in training.

## 4 GRADIENT FLOW AND NETWORK PRUNING

Gradient flow, or gradient descent with infinitesimal learning rate, is a continuous-time version of gradient descent. The evolution over time of model parameters, gradient, and loss under gradient flow can be described as follows.

$$\begin{aligned}
\text{(Parameters over time)} \quad & \frac{\partial \mathbf{\Theta}(t)}{\partial t} = -\mathbf{g}(\mathbf{\Theta}(t)); \\
\text{(Gradient over time)} \quad & \frac{\partial \mathbf{g}(\mathbf{\Theta}(t))}{\partial t} = -\mathbf{H}(\mathbf{\Theta}(t))\mathbf{g}(\mathbf{\Theta}(t)); \\
\text{(Loss over time)} \quad & \frac{\partial L(t)}{\partial t} = -\|\mathbf{g}(\mathbf{\Theta}(t))\|_2^2.
\end{aligned} \tag{6}$$

Recall that standard importance measures based on loss-preservation (Equation 3) or increase in gradient-norm (Equation 5) are derived using a first-order Taylor series approximation, making them exactly valid under the continuous scenario of gradient flow. This indicates analyzing the evolution of model parameters via gradient flow can provide useful insights into the relationships between different importance measures. To this end, we use gradient flow to develop *a general framework that relates different classes of importance measures through the norm of model parameters*. As we develop this framework, we explain the reasons why importance measures defined for pruning trained models are also highly effective when used for pruning early-on in training. In Appendix B, we further extend this framework to models trained using stochastic gradient descent, showing that up to a first-order approximation, the observations developed here are valid under SGD training too.

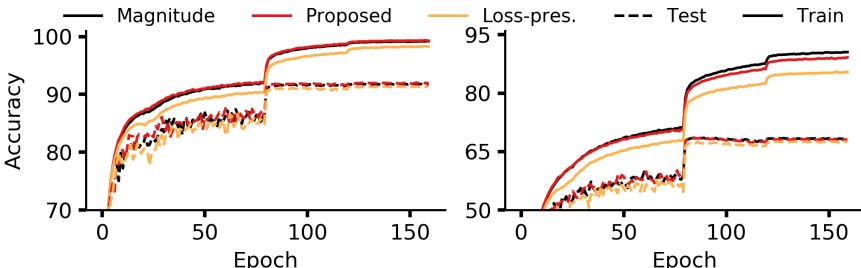

Figure 1: Train/test accuracy curves for pruned ResNet-56 models on CIFAR-10 (left) and CIFAR-100 (right) over 25 rounds. Models are pruned using magnitude-based pruning (Magnitude), the proposed extension to loss preservation (Proposed), and loss-preservation based pruning (Loss-pres.). Magnitude-based pruning converges fastest, followed by the proposed measure. Curves for other models and number of rounds are shown in Appendix F.

### 4.1 GRADIENT FLOW AND MAGNITUDE-BASED PRUNING

We first analyze the evolution of the $\ell 2$ norm of model parameters, a magnitude-based pruning measure, as a function of time under gradient flow. For a model initialized as $\boldsymbol{\Theta}(0)$, we denote parameters at time $T$ as $\boldsymbol{\Theta}(T)$. Under this notation, the distance from initialization can be related to model loss as follows:

$$\|\boldsymbol{\Theta}(T) - \boldsymbol{\Theta}(0)\|_2^2 = \left\|\int_0^T \frac{\partial \boldsymbol{\Theta}(t)}{\partial t} dt\right\|_2^2 = \left\|\int_0^T \mathbf{g}(\boldsymbol{\Theta}(t)) dt\right\|_2^2 \overset{(a)}{\leq} T \int_0^T \|\mathbf{g}(\boldsymbol{\Theta}(t)) dt\|_2^2 \, dt$$

$$\overset{(b)}{=} T \int_0^T -\frac{\partial L(t)}{\partial t} dt = T \left(L(0) - L(T)\right) \tag{7}$$

$$\implies \frac{1}{T} \|\boldsymbol{\Theta}(T) - \boldsymbol{\Theta}(0)\|_2^2 \leq L(0) - L(T),$$

where (a) follows from the application of triangle inequality for norms and the Cauchy-Schwarz inequality and (b) follows from Equation 6. The inequality above is a modification of a previously known result by Nagarajan & Kolter (2017), who show that change in model parameters, as measured by distance from initialization, is bounded by the ratio of loss at initialization to the norm of the gradient at time $T$. Frequently used initialization techniques are zero-centered with small variance (Glorot & Bengio (2010); He et al. (2015)). This reduces the impact of initialization, making $\|\boldsymbol{\Theta}(T) - \boldsymbol{\Theta}(0)\|_2^2 \approx \|\boldsymbol{\Theta}(T)\|_2^2$. In fact, calculating the importance of filters in a model using $\ell 2$ norm versus using distance from initialization, we find the two measures have an average correlation of 0.994 across training epochs, and generally prune identical parameters. We now use Equation 7 to relate magnitude-based pruning with model loss at any given time. Specifically, reorganizing Equation 7 yields

$$L(T) \leq L(0) - \frac{1}{T} \|\boldsymbol{\Theta}(T) - \boldsymbol{\Theta}(0)\|_2^2 \approx L(0) - \frac{1}{T} \|\boldsymbol{\Theta}(T)\|_2^2. \tag{8}$$

Based on this analysis, we make the following observation.

**Observation 1:** *The larger the magnitude of parameters at a particular instant, the smaller the model loss at that instant will be. If these large-magnitude parameters are preserved while pruning (instead of smaller ones), the pruned model's loss decreases faster.*

Equation 8 shows that the norm of model parameters bounds model loss at any given time. Thus, by preserving large-magnitude parameters, magnitude-based pruning enables faster reduction in loss than magnitude-agnostic techniques. For example, we later show that loss-preservation based pruning removes the most slowly changing parameters, regardless of their magnitude. Thus magnitude-based pruned models will generally converge faster than loss-preservation based pruned models.

To verify this analysis also holds under the SGD training used in practice, we train several VGG-13, MobileNet-V1, and ResNet-56 models on CIFAR-10 and CIFAR-100. Our pruning setup starts with

Table 1: Accuracy of pruned models on CIFAR-10/CIFAR-100 (over 3 seeds; base model accuracies are reported in parentheses; std. dev. $< 0.3$ for all experiments; best results are in bold, second best are underlined). Magnitude-based pruning (Mag.) and the proposed extension to loss preservation (Proposed: $\sum_{\theta_i \in \Theta_p} |\theta_i(t)||\theta_i(t)g\left(\theta_i(t)\right)|$) consistently outperform plain loss-preservation based pruning (Loss). With more rounds, the proposed measure outperforms Magnitude-based pruning too.

| Pruning Rounds | | | 1 round | | | 5 rounds | | | 25 rounds | |
|---|---|---|---|---|---|---|---|---|---|---|
| CIFAR-10 | % pruned | Mag. | Loss | Proposed | Mag. | Loss | Proposed | Mag. | Loss | Proposed |
| VGG-13 (93.1) | 75% | 92.05 | 92.01 | **92.13** | **92.32** | 91.43 | 92.29 | 92.06 | 91.53 | **92.45** |
| MobileNet (92.3) | 75% | 91.71 | 91.17 | **91.73** | 91.76 | 90.99 | **91.89** | 91.52 | 90.96 | **91.77** |
| ResNet-56 (93.1) | 60% | 91.41 | 91.09 | **91.54** | 91.80 | 91.39 | **91.88** | 91.95 | 91.47 | **92.17** |
| CIFAR-100 | % pruned | Mag. | Loss | Proposed | Mag. | Loss | Proposed | Mag. | Loss | Proposed |
| VGG-13 (69.6) | 65% | 67.89 | 68.61 | **69.01** | **69.31** | 67.88 | 68.25 | 68.31 | 68.11 | **68.93** |
| MobileNet (69.1) | 65% | 68.01 | 67.16 | **68.52** | 68.33 | 67.21 | **68.41** | 67.58 | 67.31 | **68.35** |
| ResNet-56 (71.0) | 50% | 66.92 | 66.88 | **67.10** | **68.04** | 67.11 | 67.70 | **68.75** | 67.74 | 68.62 |

randomly initialized models and uses a prune-and-train framework, where each round of pruning involves an epoch of training followed by pruning. A target amount of pruning (e.g., 75% filters) is divided evenly over a given number of pruning rounds. Throughout training, we use a small temperature value ($= 5$) to ensure smooth changes to model parameters. We provide results for 1, 5, and 25 rounds for all models and datasets. The results after 1 round, where pruning is single-shot, demonstrate that our claims are general and not an artifact of allowing the model to compensate for its lost parameters; the results after 5 and 25 rounds, where pruning is distributed over a number of rounds, demonstrate that our claims also hold when models compensate for lost parameters.

The results are shown in Table 1. Magnitude-based pruning *consistently* performs better than loss-preservation based pruning. Furthermore, train/test convergence for magnitude-based pruned models is *faster* than that for loss-preservation based pruned models, as shown in Figure 1. These results validate our claim that magnitude-based pruning results in faster converging, better-performing models when pruning early-on in training.

**Extending existing importance measures**: A fundamental understanding of existing importance measures can be exploited to extend and design new measures for pruning models early-on in training. For example, we modify loss-preservation based pruning to remove small-magnitude parameters that do not affect model loss using the following importance measure: $\sum_{\theta_i \in \Theta_p} |\theta_i(t)||\theta_i(t)g\left(\theta_i(t)\right)|$. Using $|\theta_i(t)g\left(\theta_i(t)\right)|$ to preserve loss, this measure retains training progress up to the current instant; using $|\theta_i(t)|$, this measure is biased towards removing small-magnitude parameters and should produce a model that converges faster than loss-preservation alone. These expected properties are demonstrated in Table 1 and Figure 1: the proposed measure consistently outperforms loss-preservation based pruning and often outperforms magnitude-based pruning, especially when more rounds are used (see Table 1); train/test convergence rate for models pruned using this measure are better than those for loss-preservation pruning, and competitive to those of magnitude-based pruning (see Figure 1).

## 4.2 GRADIENT FLOW AND LOSS-PRESERVATION BASED PRUNING

Loss-preservation based pruning methods use first-order Taylor decomposition to determine the impact of pruning on model loss (see Equation 3). We now show that magnitude-based and loss-preservation based pruning measures are related by an order of time-derivative.

$$\frac{\partial \|\mathbf{\Theta}(t)\|_2^2}{\partial t} = 2\mathbf{\Theta}^T(t)\frac{\partial \mathbf{\Theta}(t)}{\partial t}$$
$$\stackrel{(a)}{=} -2\mathbf{\Theta}^T(t)\mathbf{g}(\mathbf{\Theta}(t)), \tag{9}$$

where (a) follows from Equation 6. This implies that

$$\left|\frac{\partial \|\mathbf{\Theta}(t)\|_2^2}{\partial t}\right| = 2\left|\mathbf{\Theta}^T(t)\mathbf{g}(\mathbf{\Theta}(t))\right|. \tag{10}$$

Based on Equation 10, the following observations can be made.

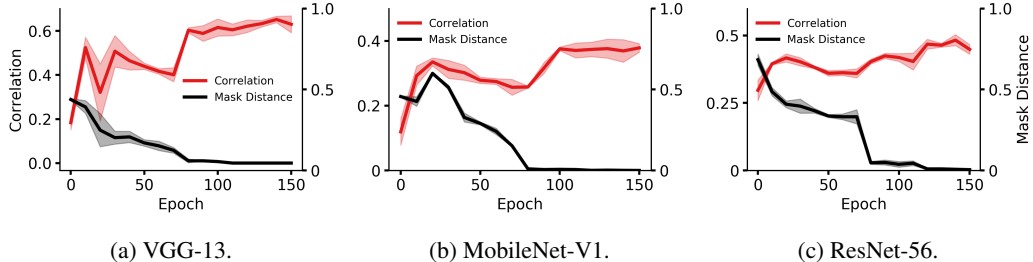

(a) VGG-13.        (b) MobileNet-V1.        (c) ResNet-56.

Figure 2: Correlation between $|\sigma\Delta\sigma|$ and loss-preservation based importance (see Equation 3) at every 10th epoch. Also plotted is the distance between pruning masks (target ratio: 20% filters), as used by You et al. (2020) to decide when to prune a model. As the distance between pruning masks over consecutive epochs reduces, $|\sigma\Delta\sigma|$ becomes more correlated with loss-preservation importance. Results for a similar experiment on Tiny-ImageNet are shown in Appendix I and in an unstructured pruning setting are shown in Section H.1.

**Observation 2:** *Up to a constant, the magnitude of time-derivative of norm of model parameters (the score for magnitude-based pruning) is equal to the importance measure used for loss-preservation (Equation 3). Further, loss-preservation corresponds to removal of the slowest changing parameters.*[2]

Equation 10 implies that loss-preservation based pruning preserves more than model loss: It also preserves the first-order dynamics of a model's evolution. *This result demonstrates that loss-preservation based importance is in fact a well-motivated measure for pruning models early-on in training.* This explains why loss-preservation based pruning has been successfully used for pruning both trained (Molchanov et al. (2017; 2019)) and untrained (Lee et al. (2019)) models. We also draw another important conclusion pertaining to pruning of trained models: *parameters that continue to change are more important to retain for preserving model loss, while parameters that have the least tendency to change are dispensable*. We highlight the relationship in Equation 10 was also noticed by Tanaka et al. (2020), but they do not study its implications in detail.

**Observation 3:** *Due to their closely related nature, when used with additional heuristics, magnitude-based importance measures preserve loss.*

Pruning methods often use additional heuristics with existing importance measures to improve their efficacy. E.g., You et al. (2020) recently showed that models become amenable to pruning early-on in training. To determine how much to train before pruning, they create a binary mask to represent filters that have low importance and are thus marked for pruning. A filter's importance is defined using the magnitude of its corresponding BatchNorm-scale parameter. If change in this binary mask is minimal over a predefined number of consecutive epochs, training is stopped and the model is pruned.

Due to the related nature of magnitude-based and loss-preservation based measures, as shown in Equation 10, we expect use of additional heuristics can unknowingly extend an importance measure to preserve model properties that it was not intentionally designed to target. To demonstrate this concretely, we analyze the implications of the "train until minimal change" heuristic by You et al., as explained above. In particular, consider a certain BatchNorm-scale parameter $\sigma$ that has a small magnitude and is thus marked for pruning. Its magnitude must continue to remain small to ensure the binary mask does not change. This implies the change in $\sigma$ over an epoch, or $\Delta\sigma$, must be small. These two constraints can be assembled in a single measure: $|\sigma\Delta\sigma|$. The continuous-time, per-iteration version of this product is $\left|\sigma\frac{\partial\sigma}{\partial t}\right|$. As shown in Equation 9 and Equation 10, $\left|\sigma\frac{\partial\sigma}{\partial t}\right|$ is the mathematical equivalent of loss-preservation based importance (Equation 3). *Therefore, when applied per iteration, removing parameters with small magnitude, under the additional heuristic of minimal change, converts magnitude-based pruning into a loss-preservation strategy.* If parameters do not change much over several iterations, this argument can further extend to change over an epoch. To confirm this, we train VGG-13, MobileNet-V1, and ResNet-56 models on CIFAR-100, record the distance between pruning masks for consecutive epochs, and calculate correlation between loss-preservation based importance and $|\sigma\Delta\sigma|$. As shown in Figure 2, as the mask distance reduces,

---

[2]This observation also implies loss preservation's importance measure does not depend on magnitude of parameters directly. We thus call loss-preservation a magnitude-agnostic technique.

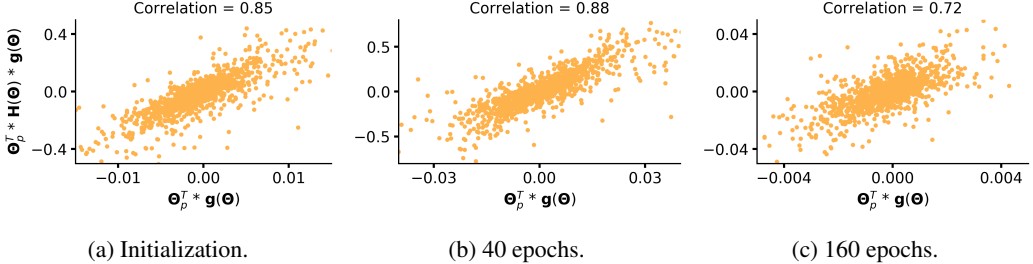

Figure 3: $\mathbf{\Theta}_p^T(t)\mathbf{H}(\mathbf{\Theta}(t))\mathbf{g}(\mathbf{\Theta}(t))$ versus $\mathbf{\Theta}_p^T(t)\mathbf{g}(\mathbf{\Theta}(t))$ for ResNet-56 models trained on CIFAR-100. Plots are shown for filters (a) at initialization, (b) after 40 epochs of training, and (c) after complete (160 epochs) training. The correlation is averaged over 3 seeds and plots are for 1 seed. As shown, the measures are highly correlated throughout model training, indicating gradient-norm increase may severely affect model loss if a partially or completely trained model is pruned using $\mathbf{\Theta}_p^T(t)\mathbf{H}(\mathbf{\Theta}(t))\mathbf{g}(\mathbf{\Theta}(t))$. Plots for other models are shown in Appendix G. Results of this experiment in an unstructured pruning setting are shown in Section H.1.

the correlation between the importance measures increases. This confirms that due to the intimate relationship between magnitude-based pruning and loss-preservation based pruning, when used with additional heuristics, magnitude-based pruning extends to preserve model loss as well.

## 4.3 GRADIENT FLOW AND GRADIENT-NORM BASED PRUNING

Having demonstrated the relationship between the first-order time derivative of the norm of model parameters and loss-preservation based pruning, we now consider the second-order time derivative of the norm of model parameters. We demonstrate its connection with GraSP (Wang et al. (2020)), a method designed for use early-on in training to increase gradient-norm using pruning.

$$\frac{\partial^2 \|\mathbf{\Theta}(t)\|_2^2}{\partial t^2} = \frac{\partial}{\partial t}\left(\frac{\partial \|\mathbf{\Theta}(t)\|_2^2}{\partial t}\right) \overset{(a)}{=} -2\frac{\partial\left(\mathbf{\Theta}^T(t)\mathbf{g}(\mathbf{\Theta}(t))\right)}{\partial t}$$

$$= -2\left(\mathbf{g}^T(\mathbf{\Theta}(t))\frac{\partial\mathbf{\Theta}(t)}{\partial t} + \mathbf{\Theta}^T(t)\frac{\partial\mathbf{g}(\mathbf{\Theta}(t))}{\partial t}\right) \quad (11)$$

$$\overset{(b)}{=} 2\left(\|\mathbf{g}(\mathbf{\Theta}(t))\|_2^2 + \mathbf{\Theta}^T(t)\mathbf{H}(\mathbf{\Theta}(t))\mathbf{g}(\mathbf{\Theta}(t))\right),$$

where (a) follows from Equation 9 and (b) follows from Equation 6. This implies that

$$\frac{\partial^2 \|\mathbf{\Theta}(t)\|_2^2}{\partial t^2} = 2\left(\|\mathbf{g}(\mathbf{\Theta}(t))\|_2^2 + \mathbf{\Theta}^T(t)\mathbf{H}(\mathbf{\Theta}(t))\mathbf{g}(\mathbf{\Theta}(t))\right). \quad (12)$$

Equation 12 shows the second-order time derivative of the norm of model parameters is linearly related to the importance measure for increase in gradient-norm (Equation 5). In particular, parameters with a negative $\mathbf{\Theta}_p^T(t)\mathbf{H}(\mathbf{\Theta}(t))\mathbf{g}(\mathbf{\Theta}(t))$ reduce the "acceleration" at which a model approaches its final solution. Thus, removing them can speedup optimization.

We now use the analysis above to demonstrate limitations to increase of gradient-norm using pruning.

**Observation 4:** *Increasing gradient-norm via pruning removes parameters that maximally increase model loss.*

Extending the "acceleration" analogy, recall that if an object has increasingly negative velocity, its acceleration will be negative as well. As shown before in Equation 9, the velocity of norm of model parameters is a constant multiple of the first-order Taylor decomposition for model loss around current parameter values. Thus, the parameter with the most negative value for $\mathbf{\Theta}_p^T(t)\mathbf{g}(\mathbf{\Theta}(t))$ is likely to also have a large, negative value for $\mathbf{\Theta}_p^T(t)\mathbf{H}(\mathbf{\Theta}(t))\mathbf{g}(\mathbf{\Theta}(t))$. From Equation 2, we observe that removing parameters with negative $\mathbf{\Theta}_p^T(t)\mathbf{g}(\mathbf{\Theta}(t))$ increases model loss with respect to the original

Table 2: Accuracy of models pruned using different variants of gradient-norm based pruning on CIFAR-10/CIFAR-100 (over 3 seeds; base model accuracies are reported in parentheses; std. dev. < 0.3 for all models, except for GraSP (T=1) with 5/25 rounds (std. dev. < 2.1); best results are in bold, second best are underlined). Gradient-norm preservation (|GraSP (T=1)|) outperforms both GraSP with large temperature (T=200) and without temperature (T=1).

| Pruning Rounds | | | 1 round | | | 5 rounds | | | 25 rounds | | |
|---|---|---|---|---|---|---|---|---|---|---|---|
| CIFAR-10 | pruned (%) | GraSP (T=200) | GraSP (T=1) | \|GraSP\| (T=1) | GraSP (T=200) | GraSP (T=1) | \|GraSP\| (T=1) | GraSP (T=200) | GraSP (T=1) | \|GraSP\| (T=1) |
| VGG-13 (93.1) | 75% | 91.62 | 91.32 | **91.92** | 90.46 | 83.77 | **91.83** | 89.57 | 77.12 | **91.75** |
| MobileNet (92.3) | 75% | 89.46 | 90.03 | **91.03** | 86.88 | 80.95 | **90.89** | 80.37 | 76.06 | **91.01** |
| ResNet-56 (93.1) | 60% | 91.01 | 90.81 | **91.21** | 90.44 | 80.15 | **91.25** | 86.23 | 10.00 | **91.63** |
| CIFAR-100 | pruned (%) | GraSP (T=200) | GraSP (T=1) | \|GraSP\| (T=1) | GraSP (T=200) | GraSP (T=1) | \|GraSP\| (T=1) | GraSP (T=200) | GraSP (T=1) | \|GraSP\| (T=1) |
| VGG-13 (69.6) | 65% | 68.51 | 65.79 | **68.64** | 66.40 | 54.56 | **68.07** | 65.18 | 42.83 | **68.13** |
| MobileNet (69.1) | 65% | 64.52 | 64.79 | **67.17** | 63.02 | 53.35 | **67.28** | 59.14 | 46.41 | **67.69** |
| ResNet-56 (71.0) | 50% | 67.09 | 67.02 | **67.15** | 66.97 | 52.98 | **67.01** | 57.55 | 1.00 | **68.02** |

model. This indicates $\mathbf{\Theta}_p^T(t)\mathbf{H}(\mathbf{\Theta}(t))\mathbf{g}(\mathbf{\Theta}(t))$ may in fact increase the gradient norm by removing parameters that maximally increase model loss.

To test this claim, we plot $\mathbf{\Theta}_p^T(t)\mathbf{H}(\mathbf{\Theta}(t))\mathbf{g}(\mathbf{\Theta}(t))$ alongside $\mathbf{\Theta}_p^T(t)\mathbf{g}(\mathbf{\Theta}(t))$ for filters of VGG-13, MobileNet-V1, and ResNet-56 models trained on CIFAR-100 at various points in training and consistently find them to be highly correlated (see Figure 3). This strong correlation confirms that *using $\mathbf{\Theta}_p^T(t)\mathbf{H}(\mathbf{\Theta}(t))\mathbf{g}(\mathbf{\Theta}(t))$ as an importance measure for network pruning increases gradient norm by removing parameters that maximally increase model loss.*

Wang et al. (2020) remark in their work that it is possible that GraSP *may* increase the gradient-norm by increasing model loss. We provide evidence and rationale illuminating this remark: we show why preserving loss and increasing gradient-norm are antithetical. To avoid this behavior, Wang et al. propose to use a large temperature value (200) before calculating the Hessian-gradient product. We now demonstrate the pitfalls of this solution and propose a more robust approach.

**Observation 5:** *Preserving gradient-norm maintains second-order model evolution dynamics and results in better-performing models than increasing gradient-norm.*

Equation 12 shows the measure $\mathbf{\Theta}_p^T(t)\mathbf{H}(\mathbf{\Theta}(t))\mathbf{g}(\mathbf{\Theta}(t))$ affects a model's second-order evolution dynamics. The analysis in Observation 4 shows $\mathbf{\Theta}_p^T(t)\mathbf{H}(\mathbf{\Theta}(t))\mathbf{g}(\mathbf{\Theta}(t))$ and $\mathbf{\Theta}_p^T(t)\mathbf{g}(\mathbf{\Theta}(t))$ are highly correlated. These results together imply that *preserving* gradient-norm $\left(\left|\mathbf{\Theta}_p^T(t)\mathbf{H}(\mathbf{\Theta}(t))\mathbf{g}(\mathbf{\Theta}(t))\right|\right)$ should preserve both second-order model evolution dynamics and model loss. Since the correlation with loss-preservation based importance is not perfect, this strategy is only an approximation of loss-preservation. However, it is capable of avoiding increase in model loss due to pruning.

To demonstrate the efficacy of gradient-norm preservation and the effects of increase in model loss on GraSP, we prune VGG-13, MobileNet-V1, and ResNet-56 models trained on CIFAR-10 and CIFAR-100. The results are shown in Table 2 and lead to the following conclusions. (i) When a single round of pruning is performed, the accuracy of GraSP is essentially independent of temperature. This implies that using temperature may be unnecessary if the model is close to initialization. (ii) In a few epochs of training, when reduction in loss can be attributed to training of earlier layers (Raghu et al. (2017)), GraSP without large temperature chooses to prune earlier layers aggressively (see Section G.2 for layer-wise pruning ratios). This permanently damages the model and thus the accuracy of low-temperature GraSP decreases substantially with increasing training epochs. (iii) At high temperatures, the performance of pruned models is more robust to the number of rounds and pruning of earlier layers is reduced. However, reduction in accuracy is still significant. (iv) These behaviors are mitigated by using the proposed alternative: gradient-norm preservation $\left(\left|\mathbf{\Theta}_p^T(t)\mathbf{H}(\mathbf{\Theta}(t))\mathbf{g}(\mathbf{\Theta}(t))\right|\right)$. Since this measure preserves the gradient-norm and is correlated to loss-preservation, it focuses on ensuring the model's second-order training dynamics and loss remain the same, consistently resulting in the best performance.

## 5 Conclusion

In this paper, we revisit importance measures designed for pruning trained models to better justify their use early-on in training. Developing a general framework that relates these measures through the norm of model parameters, we analyze what properties of a parameter make it more dispensable according to each measure. This enables us to show that from the lens of model evolution, use of magnitude- and loss-preservation based measures is well-justified early-on in training. More specifically, by preserving parameters that enable fast convergence, magnitude-based pruning generally outperforms magnitude-agnostic methods. By removing parameters that have the least tendency to change, loss-preservation based pruning preserves first-order model evolution dynamics and is well-justified for pruning models early-on in training. We also explore implications of the intimate relationship between magnitude-based pruning and loss-preservation based pruning, demonstrating that one can evolve into the other as training proceeds. Finally, we analyze gradient-norm based pruning and show that it is linearly related to second-order model evolution dynamics. Due to this relationship, we find that increasing gradient-norm via pruning corresponds to removing parameters that maximally increase model loss. Since such parameters are concentrated in the initial layers early-on in training, this method can permanently damage a model's initial layers and undermine its ability to learn from later training. To mitigate this problem, we show the more robust approach is to prune parameters that preserve gradient norm, thus preserving a model's second-order evolution dynamics while pruning. In conclusion, our work shows the use of an importance measure for pruning models early-on in training is difficult to justify unless the measure's relationship with the evolution of a model's parameters over time is established. More generally, we believe new importance measures that specifically focus on pruning early-on should be directly motivated by a model's evolution dynamics.

## 6 Acknowledgements

We thank Puja Trivedi, Daniel Kunin, Hidenori Tanaka, and Chaoqi Wang for several helpful discussions during the course of this project. We also thank Puja Trivedi and Karan Desai for providing useful comments in the drafting of this paper.

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

# A    ORGANIZATION

The appendix is organized as follows:

- Appendix B: Extends our gradient flow framework towards SGD based training.

- Appendix C: Details the setup for training base models.

- Appendix D: Details the setup for training pruned models.

- Appendix E: Random/Uniform pruning results on CIFAR-100.

- Appendix F: Train/Test curves for magnitude-based and loss-preservation based pruned models.

- Appendix G: More Results on Gradient-norm Based Pruning.

    Section G.1: Scatter plots demonstrating correlation between $\mathbf{\Theta}_p^T(t)\mathbf{H}(\mathbf{\Theta}(t))\mathbf{g}(\mathbf{\Theta}(t))$ and $\mathbf{\Theta}_p^T(t)\mathbf{g}(\mathbf{\Theta}(t))$.

    Section G.2: Provides layer-wise pruning ratios for models pruned using gradient-norm based pruning variants.

    Section G.3: Train/Test curves for gradient-norm based variants used in the main paper.

- Appendix H: Provides empirical verification for observations 2–4 in an unstructured pruning setup.

- Appendix I: Provides empirical verification for observations 2–4 on Tiny-ImageNet.

# B    EXTENDING THE FRAMEWORK TO SGD TRAINING

The main paper focuses on using gradient flow to establish the relationship between frequently used importance measures for network pruning and evolution of model parameters. In practice, however, deep neural networks are trained using SGD (stochastic gradient descent) or its variants (e.g., SGD with momentum). *This section serves to demonstrate that up to a first order approximation, our observations based upon gradient flow are theoretically valid for SGD training too.* Our analysis is based on the fact that when an importance measure is employed, most pruning frameworks stop model training and use several mini-batches of training data to determine an approximation for gradient/Hessian. This enables us to use an expectation operation over mini-batches of data, making a theoretical analysis tractable.

## B.1    NOTATIONS AND SGD UPDATES

**Notations:** We first establish new notations for this section. For completeness, notations used in the main paper are recapped as well:
$\mathbf{\Theta}(t)$ denotes model parameters at time $t$. $\eta$ denotes the learning rate at which stochastic gradient descent trains a model. $\mathbf{d}(\mathbf{\Theta}(t); X_i)$ denotes the estimate of gradient of loss and $\mathbf{H}(\mathbf{\Theta}(t); X_i)$ denotes the estimate of Hessian of loss with respect to model parameters at time $t$, as calculated using the $i^{th}$ mini-batch ($X_i$). $\mathbf{g}(\mathbf{\Theta}(t))$ denotes the full-batch gradient of loss and $\mathbf{H}(\mathbf{\Theta}(t))$ denotes Hessian of loss with respect to model parameters at time $t$.

**SGD Update:** Under stochastic gradient descent, a random mini-batch of data is sampled amongst all mini-batches without replacement. For example, if mini-batch $X_i$ is sampled, a gradient approximation based on its data elements will be used to update model parameters at the current iteration. Therefore, the model parameters evolve as follows:

$$\mathbf{\Theta}(t) \rightarrow \mathbf{\Theta}(t) - \eta\mathbf{d}(\mathbf{\Theta}(t); X_i). \tag{13}$$

## B.2 Change in norm of model parameters: $\Delta \|\Theta(t)\|_2^2$

We first use SGD to analyze change in norm of model parameters, denoted as $\Delta \|\Theta(t)\|_2^2$. Assume mini-batch $X_i$ is used to update model parameters at the current iteration.

$$
\begin{aligned}
\Delta \|\Theta(t)\|_2^2 &= \|\Theta(t) - \eta \mathbf{d}(\Theta(t); X_i)\|_2^2 - \|\Theta(t)\|_2^2 \\
&= \|\Theta(t)\|_2^2 - 2\eta \, \Theta(t)^T \mathbf{d}(\Theta(t); X_i) + \eta^2 \|\mathbf{d}(\Theta(t); X_i)\|_2^2 - \|\Theta(t)\|_2^2 \\
&= -2\eta \, \Theta(t)^T \mathbf{d}(\Theta(t); X_i) + \mathcal{O}(\eta^2) \\
&\approx -2\eta \, \Theta(t)^T \mathbf{d}(\Theta(t); X_i),
\end{aligned}
\tag{14}
$$

where we ignored terms with higher order products of learning rate $\eta$. Smaller the learning rate, the better this approximation will be.

As mentioned before, to compute importance estimates for network pruning, most pruning frameworks stop the training process and compute estimates for gradient/Hessian using several mini-batches of data. To account for this, we can take expectation over the mini-batch sampling process. For example, if input data is denoted as $X$, under expectation, Equation 14 changes as follows:

$$
\begin{aligned}
E_{X_i \sim X} \left[ \Delta \|\Theta(t)\|_2^2 \right] &= E_{X_i \sim X} \left[ -2\eta \, \Theta(t)^T \mathbf{d}(\Theta(t); X_i) \right] \\
&= -2\eta \, \Theta(t)^T E_{X_i \sim X} \left[ \mathbf{d}(\Theta(t); X_i) \right] \\
&= -2\eta \, \Theta(t)^T \mathbf{g}(\Theta(t)).
\end{aligned}
\tag{15}
$$

$$
\implies E_{X_i \sim X} \left[ \frac{\Delta \|\Theta(t)\|_2^2}{\eta} \right] = -2\Theta(t)^T \mathbf{g}(\Theta(t)).
$$

As the main paper shows (see Equation 9), under gradient flow, the norm of model parameters evolves as follows:

$$
\frac{\partial \|\Theta(t)\|_2^2}{\partial t} = -2\Theta(t)^T \mathbf{g}(\Theta(t)).
\tag{16}
$$

Therefore, using Equation 15 and Equation 16, it can be evidently seen that up to a first-order approximation, the expected rate at which norm of model parameters evolves under SGD is exactly the same as that under gradient flow. In the main paper, we use this relationship to conclude Observations 2 and 3. Thus, our analysis in this section indicates *Observations 2, 3 are in fact valid approximations under SGD training as well.*

## B.3 Change in change of norm of model parameters: $\Delta^2 \|\Theta(t)\|_2^2$

We now determine the expected rate at which change in norm of model parameters itself changes under SGD. To this end, we will calculate $\Delta^2 \|\Theta(t)\|_2^2 = \Delta(\Delta \|\Theta(t)\|_2^2)$.

Equation 14 shows that when mini-batch $X_i$ is used to take an optimization step, up to a first-order approximation, the change in norm of model parameters ($\Delta \|\Theta(t)\|_2^2$) is as follows:

$$
\Delta \|\Theta(t)\|_2^2 = -2\eta \, \Theta(t)^T \mathbf{d}(\Theta(t); X_i).
\tag{17}
$$

After the optimization step based on mini-batch $X_i$ has been completed, the model parameters change from $\Theta(t)$ to $\Theta(t) - \eta \mathbf{d}(\Theta(t); X_i)$. Assume that a new mini-batch $X_j$ is sampled for the next optimization step–i.e., the next gradient estimate is $\mathbf{d}(\Theta(t) - \eta \mathbf{d}(\Theta(t); X_i); X_j)$. To relate this new gradient estimate with the previous one, we use a first-order Taylor approximation for model gradient based on mini-batch $X_j$:

$$
\mathbf{d}(\Theta(t) - \eta \mathbf{d}(\Theta(t); X_i); X_j) = \mathbf{d}(\Theta(t); X_j) - \eta \mathbf{H}(\Theta(t); X_j) \mathbf{d}(\Theta(t); X_i).
\tag{18}
$$

Note that in the above equation, the Hessian estimate is based on mini-batch $X_j$ only. Using this relationship, the change in norm of model parameters can be approximated as follows:

$$
\begin{aligned}
\Delta \left\| \Theta(t) - \eta \mathbf{d}(\Theta(t); X_i) \right\|_2^2 &= -2\eta \left( \Theta(t) - \eta \mathbf{d}\left(\Theta(t); X_i\right) \right)^T \mathbf{d}\left( \Theta(t) - \eta \mathbf{d}\left(\Theta(t); X_i\right); X_j \right) \\
&= -2\eta \left( \Theta(t) - \eta \mathbf{d}\left(\Theta(t); X_i\right) \right)^T \left( \mathbf{d}\left(\Theta(t); X_j\right) - \eta \mathbf{H}(\Theta(t); X_j)\mathbf{d}\left(\Theta(t); X_i\right) \right) \\
&= -2\eta\, \Theta(t)^T \mathbf{d}\left(\Theta(t); X_j\right) + 2\eta^2\, \Theta(t)^T \mathbf{H}(\Theta(t); X_j)\mathbf{d}\left(\Theta(t); X_i\right) \\
&\quad + 2\eta^2\, \mathbf{d}\left(\Theta(t); X_i\right)^T \mathbf{d}\left(\Theta(t); X_j\right) + \mathcal{O}(\eta^3) \\
&\approx -2\eta\, \Theta(t)^T \mathbf{d}\left(\Theta(t); X_j\right) + 2\eta^2\, \Theta(t)^T \mathbf{H}(\Theta(t); X_j)\mathbf{d}\left(\Theta(t); X_i\right) \\
&\quad + 2\eta^2\, \mathbf{d}\left(\Theta(t); X_i\right)^T \mathbf{d}\left(\Theta(t); X_j\right),
\end{aligned}
\tag{19}
$$

where we ignore terms of the order of $\mathcal{O}(\eta^3)$. This above result can now be used to calculate $\Delta^2 \left\| \Theta(t) \right\|_2^2$, our desired quantity.

$$
\begin{aligned}
\Delta^2 \left\| \Theta(t) \right\|_2^2 &= \Delta \left\| \Theta(t) - \eta \mathbf{d}(\Theta(t); X_i) \right\|_2^2 - \Delta \left\| \Theta(t) \right\|_2^2 \\
&= [-2\eta\, \Theta(t)^T \mathbf{d}\left(\Theta(t); X_j\right) + 2\eta^2\, \Theta(t)^T \mathbf{H}(\Theta(t); X_j)\mathbf{d}\left(\Theta(t); X_i\right) \\
&\quad + 2\eta^2\, \mathbf{d}\left(\Theta(t); X_i\right)^T \mathbf{d}\left(\Theta(t); X_j\right)] - [-2\eta\, \Theta(t)^T \mathbf{d}\left(\Theta(t); X_i\right)] \\
&= -2\eta\, \Theta(t)^T \left( \mathbf{d}\left(\Theta(t); X_j\right) - \mathbf{d}\left(\Theta(t); X_i\right) \right) + 2\eta^2\, \Theta(t)^T \mathbf{H}(\Theta(t); X_j)\mathbf{d}\left(\Theta(t); X_i\right) \\
&\quad + 2\eta^2\, \mathbf{d}\left(\Theta(t); X_i\right)^T \mathbf{d}\left(\Theta(t); X_j\right).
\end{aligned}
\tag{20}
$$

We again use the fact that importance estimates for network pruning are estimated by stopping model training and computing gradient/Hessian estimates over several mini-batches of data. We denote this through an expectation over input data $X$, where our random variables are the mini-batches $X_i$ and $X_j$. As mini-batches are sampled independently, we use the result that expectation of product of their functions can be independently evaluated: i.e., $E_{X_i, X_j \sim X}[f(X_i)g(X_j)] = E_{X_i \sim X}[f(X_i)]E_{X_j \sim X}[g(X_j)]$.

$$
\begin{aligned}
\implies E_{X_i, X_j \sim X}\left[ \Delta^2 \left\| \Theta(t) \right\|_2^2 \right] &= -2\eta\, E_{X_i, X_j \sim X}\left[ \Theta(t)^T \left( \mathbf{d}\left(\Theta(t); X_j\right) - \mathbf{d}\left(\Theta(t); X_i\right) \right) \right] \\
&\quad + 2\eta^2\, E_{X_i, X_j \sim X}\left[ \Theta(t)^T \mathbf{H}(\Theta(t); X_j)\mathbf{d}\left(\Theta(t); X_i\right) \right] \\
&\quad + 2\eta^2\, E_{X_i, X_j \sim X}\left[ \mathbf{d}\left(\Theta(t); X_i\right)^T \mathbf{d}\left(\Theta(t); X_j\right) \right] \\
&= -2\eta\, \Theta(t)^T \left( E_{X_j \sim X}[(\mathbf{d}\left(\Theta(t); X_j\right)] - E_{X_i \sim X}[\mathbf{d}\left(\Theta(t); X_i\right)]) \right) \\
&\quad + 2\eta^2\, \Theta(t)^T E_{X_j \sim X}[\mathbf{H}(\Theta(t); X_j)]\, E_{X_i \sim X}[\mathbf{d}\left(\Theta(t); X_i\right)] \\
&\quad + 2\eta^2\, E_{X_i \sim X}[\mathbf{d}\left(\Theta(t); X_i\right)]^T E_{X_j \sim X}[\mathbf{d}\left(\Theta(t); X_j\right)] \\
&= -2\eta\, \Theta(t)^T \left( \mathbf{g}(\Theta(t)) - \mathbf{g}(\Theta(t)) \right) + 2\eta^2\, \Theta(t)^T \mathbf{H}(\Theta(t))\mathbf{g}(\Theta(t)) \\
&\quad + 2\eta^2\, \mathbf{g}(\Theta(t))^T \mathbf{g}(\Theta(t)) \\
&= 2\eta^2 \left( \left\| \mathbf{g}(\Theta(t)) \right\|^2 + \Theta(t)^T \mathbf{H}(\Theta(t))\mathbf{g}(\Theta(t)) \right).
\end{aligned}
\tag{21}
$$

$$
\implies E_{X_i, X_j \sim X}\left[ \frac{\Delta^2 \left\| \Theta(t) \right\|_2^2}{\eta^2} \right] = 2 \left( \left\| \mathbf{g}(\Theta(t)) \right\|^2 + \Theta(t)^T \mathbf{H}(\Theta(t))\mathbf{g}(\Theta(t)) \right).
\tag{22}
$$

As shown in the main paper (see Equation 12), under gradient flow, the rate at which change in norm of model parameters changes is as follows:

$$
\implies \frac{\partial^2 \left\| \Theta(t) \right\|_2^2}{\partial t^2} = 2 \left( \left\| \mathbf{g}(\Theta(t)) \right\|^2 + \Theta(t)^T \mathbf{H}(\Theta(t))\mathbf{g}(\Theta(t)) \right).
\tag{23}
$$

As can be evidently seen in Equation 22 and Equation 23, up to a first-order approximation, the expected rate at which change of norm of model parameters changes under SGD is exactly the same

as that under gradient flow. In the main paper, we use this relationship to conclude Observations 4 and 5. Thus, our analysis in this section indicates *Observations 4, 5 are in fact valid approximations under SGD training as well.*

Overall, in this section, we showed that the relationships between commonly used measures for network pruning and their expected impact on model parameters is the same for both SGD training and gradient flow, up to a first-order approximation.

## C  TRAINING BASE MODELS

We analyze models trained on CIFAR-10 and CIFAR-100 datasets. All models are trained with the exact same setup. In all our experiments, we average results across 3 seeds.

Since pruning has a highly practical motivation, we argue analyzing a variety of models that capture different architecture classes is important. To this end, we use VGG-13 (a vanilla CNN), MobileNet-V1 (a low-redundancy CNN designed specifically for efficient computation), and ResNet-56 (a residual model to analyze effects of pruning under skip connections). The final setup is as follows:

- Optimizer: SGD,
- Momentum: 0.9,
- Weight decay: 0.0001,
- Learning rate schedule: $(0.1, 0.01, 0.001)$,
- Number of epochs for each learning rate: $(80, 40, 40)$,
- Batch Size: 128.

## D  TRAINING PRUNED MODELS

The general setup for training pruned models is the same as that for training base models. However, for a fair comparison, if the prune-and-train framework takes $n$ rounds, we subtract $n$ epochs from the number of epochs alloted to the highest learning rate. This ensures same amount of training budget for all models and pruning strategies. Note that even if this subtraction is not performed, the results do not vary much, as the model already converges (see Appendix F for train/test curves). The final setup is as follows.

- Optimizer: SGD,
- Momentum: 0.9,
- Weight decay: 0.0001,
- Learning rate schedule: $(0.1, 0.01, 0.001)$,
- Number of epochs for each learning rate: $(80 - \text{number of pruning rounds}, 40, 40)$,
- Batch size: 128.

Since we prune models in a structured manner, our target pruning ratios are chosen in terms of the number of filters to be pruned. A rough translation in terms of parameters follows:

- VGG-13: 75% filters ($\sim$94% parameters) for CIFAR-10; 65% filters ($\sim$89% parameters) for CIFAR-100,
- MobileNet-V1: 75% filters ($\sim$94% parameters) for CIFAR-10; 65% filters ($\sim$89% parameters) for CIFAR-100,
- ResNet-56: 60% filters ($\sim$88% parameters) for CIFAR-10; 50% filters ($\sim$83% parameters) for CIFAR-100.

The percentage of pruned parameters is much larger than the percentage of pruned filters because generally filters from deeper layers are pruned more heavily. These filters form the bulk of a model's parameters, thus resulting in high parameter percentages.

## E    RANDOM/UNIFORM PRUNING

In this section, we compare various pruning measures with random and uniform pruning under the same experimental setup as used in the main paper (see Appendix D). For random pruning, we assign a random score between 0–1, with equal probability, to each filter. For uniform pruning, we decide a target amount of percentage pruning and remove that much percentage filters from all layers. For example, if 50% of the model is to be pruned, we remove half the filters from each layer randomly.

If which importance measure is used for pruning a model in structured pruning settings is not an important choice, both random and uniform pruning schemes should perform competitively with standardly used importance measures for network pruning. However, as we show in Table 3, random and uniform pruning perform much worse than the standardly used metrics for pruning.

Table 3: Reduction in test accuracy for CIFAR-100 models pruned randomly, uniformly, and using the importance measures analyzed in the main paper. Test accuracies of base models are reported in the table. [†]When pruning is distributed over several rounds, it often leads to small amounts of pruning in a given round. Since only integer number of filters can be pruned, a small ratio will result in no pruning at a layer with small number of filters and other layers will be pruned more to compensate for the same. While this issue does not arise in VGG-13 and MobileNet-V1 models, for ResNet-56, which has several layers with only 16 filters per layer, often very minimal amounts of pruning takes place. Due to this, later layers are pruned more aggressively than earlier layers, resulting in poor performance in both 5 and 25 rounds of random/uniform pruning for ResNet-56.

| VGG-13 (65% pruned) | Base Model | Random | Uniform | Mag. Based | Loss Based | Proposed Extension | GraSP (T=200) | GraSP (T=1) | \|GraSP\| (T=1) |
|---|---|---|---|---|---|---|---|---|---|
| 1 round | 69.62 | -5.71 | -5.60 | -1.73 | -1.01 | -0.61 | -1.11 | -3.83 | -0.96 |
| 5 rounds | | -5.49 | -4.57 | -0.31 | -1.74 | -1.37 | -3.22 | -15.06 | -1.53 |
| 25 rounds | | -4.55 | -3.19 | -1.31 | -0.51 | -0.69 | -4.44 | -26.79 | -1.48 |
| MobileNet-V1 (65% pruned) | Base Model | Random | Uniform | Mag. Based | Loss Based | Proposed Extension | GraSP (T=200) | GraSP (T=1) | \|GraSP\| (T=1) |
| 1 round | 69.10 | -6.54 | -9.71 | -1.09 | -1.94 | -0.58 | -4.58 | -4.31 | -1.93 |
| 5 rounds | | -6.55 | -4.66 | -0.77 | -1.89 | -0.69 | -6.08 | -15.35 | -1.82 |
| 25 rounds | | -5.59 | -3.56 | -1.52 | -1.79 | -0.75 | -9.96 | -22.61 | -1.41 |
| ResNet-56 (50% pruned) | Base Model | Random | Uniform | Mag. Based | Loss Based | Proposed Extension | GraSP (T=200) | GraSP (T=1) | \|GraSP\| (T=1) |
| 1 round | 71.01 | -4.00 | -3.42 | -4.09 | -4.13 | -3.91 | -3.92 | -3.99 | -3.86 |
| 5 rounds | | -9.05[†] | -7.58[†] | -2.97 | -3.90 | -3.31 | -4.04 | -18.03 | -4.00 |
| 25 rounds | | -27.54[†] | -70.01[†] | -2.26 | -3.27 | -2.39 | -13.46 | -70.01 | -2.99 |

## F    TRAIN/TEST PLOTS

This section further demonstrates that magnitude based pruned models converge faster than loss-preservation based pruned models. For magnitude-based pruning, the importance is defined as $\ell 2$ norm of a filter; for loss-preservation, the importance is defined as a variant of SNIP (Lee et al. (2019)) applied to entire filter. The plots show train/test curves for VGG-13, MobileNet-V1, and ResNet-56 models for 1, 5, and 25 pruning rounds each. Also plotted are train/test curves for the proposed extension to loss-preservation which intentionally biases loss-preservation towards removing small magnitude parameters (see Section 4.1 for more details).

## F.1 VGG-13

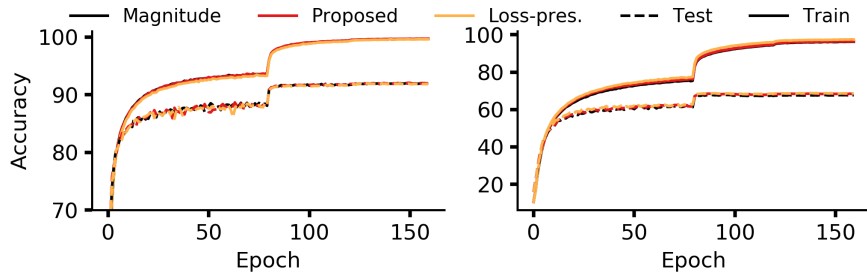

Figure 4: VGG-13: 1 round of pruning. CIFAR-10 (left); CIFAR-100 (right).

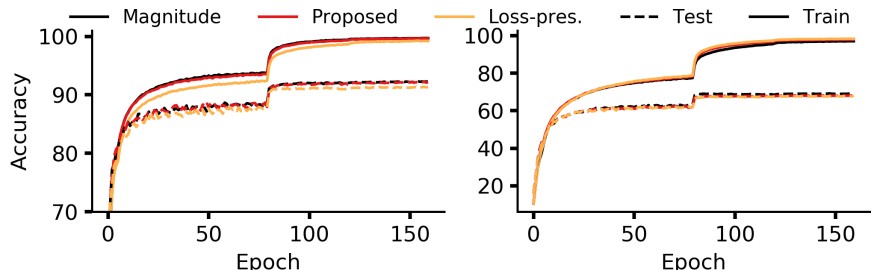

Figure 5: VGG-13: 5 rounds of pruning. CIFAR-10 (left); CIFAR-100 (right).

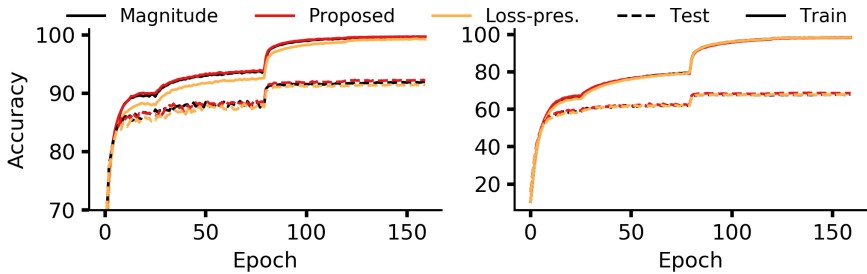

Figure 6: VGG-13: 25 rounds of pruning. CIFAR-10 (left); CIFAR-100 (right).

## F.2 MOBILENET-V1

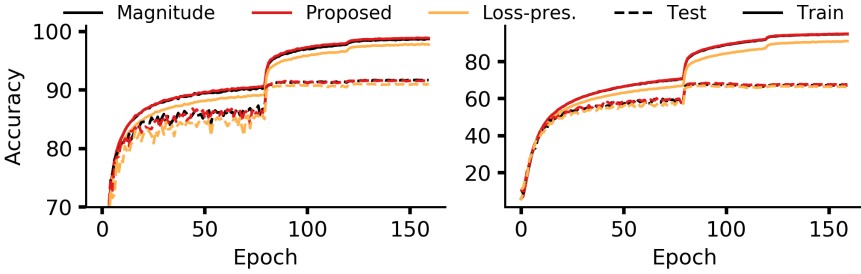

Figure 7: MobileNet-V1: 1 round of pruning. CIFAR-10 (left); CIFAR-100 (right).

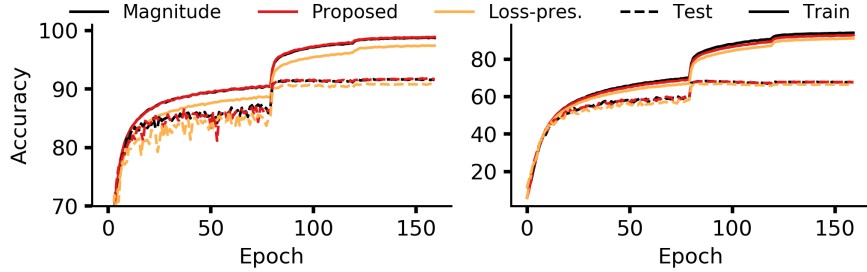

Figure 8: MobileNet-V1: 5 rounds of pruning. CIFAR-10 (left); CIFAR-100 (right).

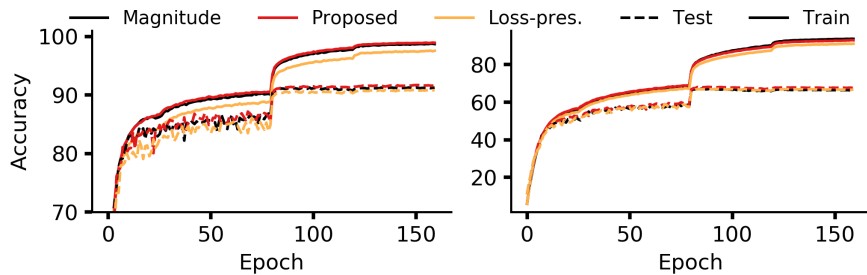

Figure 9: MobileNet-V1: 25 rounds of pruning. CIFAR-10 (left); CIFAR-100 (right).

## F.3 RESNET-56

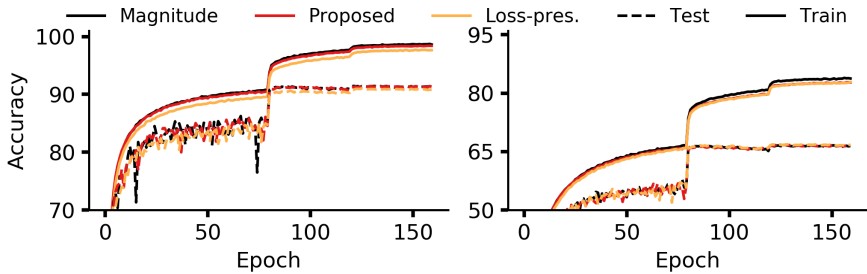

Figure 10: ResNet-56: 1 round of pruning. CIFAR-10 (left); CIFAR-100 (right).

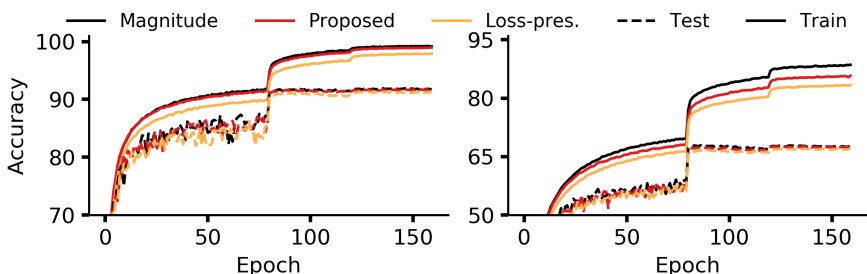

Figure 11: ResNet-56: 5 rounds of pruning. CIFAR-10 (left); CIFAR-100 (right).

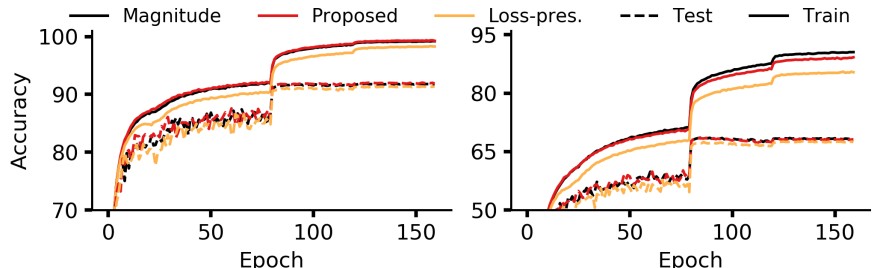

Figure 12: ResNet-56: 25 rounds of pruning. CIFAR-10 (left); CIFAR-100 (right).

## G   MORE RESULTS ON GRADIENT-NORM BASED PRUNING

This section provides further results on gradient-norm based pruning. The general implementation of these measures requires the calculation of Hessian-gradient products. Similar to the implementation by Wang et al. (2020), we define a constant amount of memory that stores randomly selected samples for all classes. For the original GraSP variant that increases gradient norm, we follow the original implementation and use a temperature of 200 during the calculation of Hessian-gradient product. This involves division of the model output (i.e., the logits vector) by a constant T (= 200) before using softmax for classification (see Hinton et al. (2015) for further details). For the GraSP variant without large temperature, this temperature value is brought down to 1. Finally, for the gradient norm preservation measure, the GraSP variant without large temperature is used alongside an absolute value operator to remove parameters that least change gradient norm (i.e., least affect second-order model evolution dynamics).

### G.1   SCATTER PLOTS FOR $\Theta_p^T(t)\mathbf{H}(\Theta(t))\mathbf{g}(\Theta(t))$ VS. $\Theta_p^T(t)\mathbf{g}(\Theta(t))$

This subsection provides scatter plots demonstrating the highly correlated nature of $\Theta_p^T(t)\mathbf{H}(\Theta(t))\mathbf{g}(\Theta(t))$ and $\Theta_p^T(t)\mathbf{g}(\Theta(t))$. The correlation is averaged over 3 seeds and plots are for 1 seed. As can be seen in the plots, the measures are highly correlated throughout model training, indicating gradient-norm increase may severely affect model loss if a partially trained or completely trained model is pruned using $\Theta_p^T(t)\mathbf{H}(\Theta(t))\mathbf{g}(\Theta(t))$.

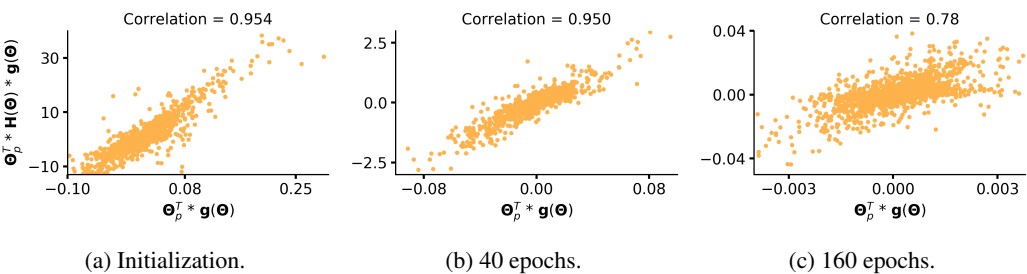

(a) Initialization.    (b) 40 epochs.    (c) 160 epochs.

Figure 13: $\Theta_p^T(t)\mathbf{H}(\Theta(t))\mathbf{g}(\Theta(t))$ versus $\Theta_p^T(t)\mathbf{g}(\Theta(t))$ for VGG-13 models trained on CIFAR-100. Plots are shown for filters (a) at initialization, (b) after 40 epochs of training, and (c) after complete (160 epochs) training.

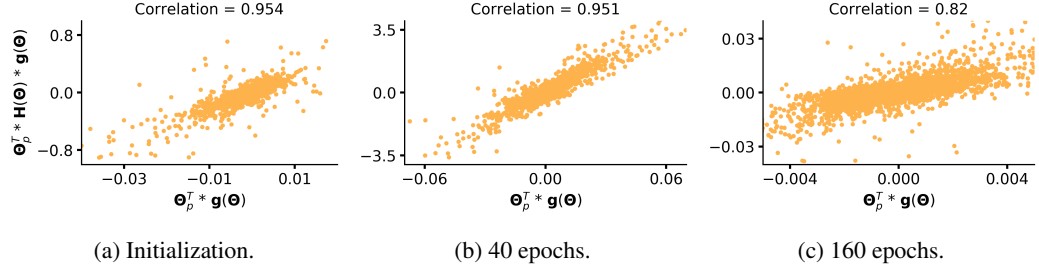

(a) Initialization.     (b) 40 epochs.     (c) 160 epochs.

Figure 14: $\mathbf{\Theta}_p^T(t)\mathbf{H}(\mathbf{\Theta}(t))\mathbf{g}(\mathbf{\Theta}(t))$ versus $\mathbf{\Theta}_p^T(t)\mathbf{g}(\mathbf{\Theta}(t))$ for MobileNet-V1 models trained on CIFAR-100. Plots are shown for filters (a) at initialization, (b) after 40 epochs of training, and (c) after complete (160 epochs) training.

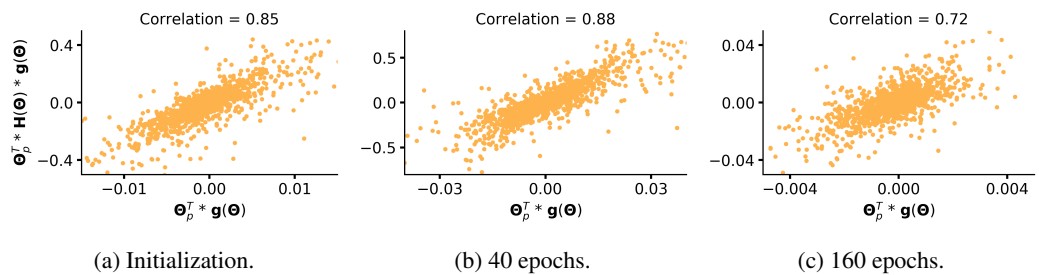

(a) Initialization.     (b) 40 epochs.     (c) 160 epochs.

Figure 15: $\mathbf{\Theta}_p^T(t)\mathbf{H}(\mathbf{\Theta}(t))\mathbf{g}(\mathbf{\Theta}(t))$ versus $\mathbf{\Theta}_p^T(t)\mathbf{g}(\mathbf{\Theta}(t))$ for ResNet-56 models trained on CIFAR-100. Plots are shown for filters (a) at initialization, (b) after 40 epochs of training, and (c) after complete (160 epochs) training.

## G.2 Layer-wise Pruning Ratios

In Table 2, we showed that in a few epochs of training, when reduction in loss can be attributed to training of earlier layers (Raghu et al. (2017)), GraSP without large temperature chooses to prune earlier layers aggressively. This failure mode is an inherent tendency of GraSP, since GraSP assigns least importance to parameters whose removal maximally increases model loss (see Observation 5 in Section 4.3).

To further elaborate on this comment, we first provide layer-wise pruning ratios of models pruned using different variants of gradient-norm based pruning–i.e., GraSP with large temperature (GraSP (T=200)), Gradient norm preservation (|GraSP|), and GraSP without large temperature (GraSP (T=1)). Figure 16 provides results for VGG-13 models; Figure 17 provides results for MobileNet-V1 models; and Figure 18, Figure 19, and Figure 20 provide results for block-1, block-2, and block-3 of ResNet-56 models, respectively. For sequential architectures (VGG-13 and MobileNet-V1), we can vividly see earlier layers are pruned aggressively. For ResNet-56, we find that in any given residual module, the first convolutional layer is pruned a lot more. This holds well with our claim that GraSP removes parameters which maximally increase the model loss, for removing parameters from the first convolutional layer in a residual module will completely block signal propagation for that module. The remaining pruning budget can then be used to remove parameters from the first convolutional layer of other residual modules, instead of focusing on the Layer-2 of the same module. Note that this behavior becomes more explicit as several rounds of pruning are used and invariably earlier Layer-1's in earlier blocks are pruned more.

Overall, across all models, GraSP without large temperature prunes earlier layers very aggressively. This permanently damages the model and thus the accuracy of low-temperature GraSP is much lower than other variants of gradient-norm based pruning. Even for GraSP with large temperature, when more rounds of pruning are used, use of large temperature is unable to satisfactorily avoid aggressive pruning in earlier layers (e.g., see Figure 17c). Only Gradient norm preservation is able to avoid this failure mode, and does so across all models and settings.

From a more theoretical standpoint, one can again employ gradient flow to better understand this behavior. In particular, Equation 6 shows model loss ($L(t)$) over time reduces in proportion to negative of the gradient norm. For example, if a model has $N$ layers and $\mathbf{\Theta}_n(t)$, $\mathbf{g}(\mathbf{\Theta}_n(t))$ denote parameters, gradient of the $n^{th}$ layer, then the model loss reduces at the following rate:

$$\frac{\partial L(t)}{\partial t} = -\|\mathbf{g}(\mathbf{\Theta}(t))\|_2^2 = -\sum_{n=1}^{N}\|\mathbf{g}(\mathbf{\Theta}_n(t))\|_2^2.$$ (24)

This implies, at any given time, if a layer has a higher gradient norm, it contributes more to reduction in loss. This makes parameters of that layer an appropriate candidate for removal by GraSP.

We plot the layer-wise gradient norm of different models trained on CIFAR-100 over time (see Figure 21). As can be evidently seen, gradient norm is highest for the earlier layers in the beginning. Later in training, especially after the learning rate is dropped ($80^{th}$ epoch), it is difficult to discern which part of the model has a higher gradient norm. Overall, this experiment provides further evidence that early-on in training, reduction in model loss can be majorly attributed to the optimization of parameters in the earlier layers, as noted by Raghu et al. (2017) through the use of SVCCA. This leads to aggressive pruning of earlier layers when GraSP is used as an importance measure. Gradient norm preservation, however, is able to avoid this failure mode.

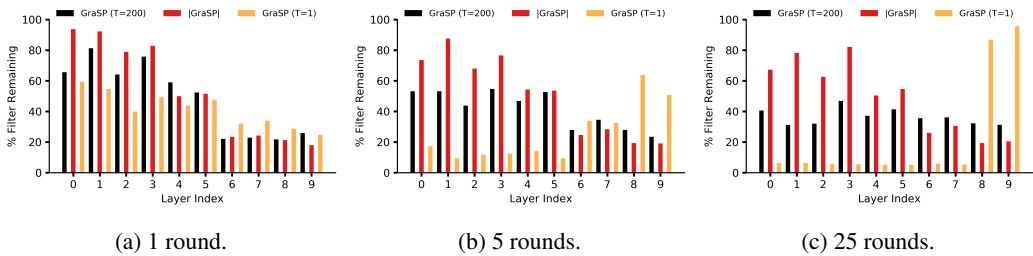

(a) 1 round.     (b) 5 rounds.     (c) 25 rounds.

Figure 16: Layer-wise pruning ratios for VGG-13 with a target ratio of 65% pruning. Results are provided for pruning over (a) 1 round, (b) 5 rounds, and (c) 25 rounds. When the model is allowed to train even slightly, GraSP without temperature prunes earlier layers very aggressively.

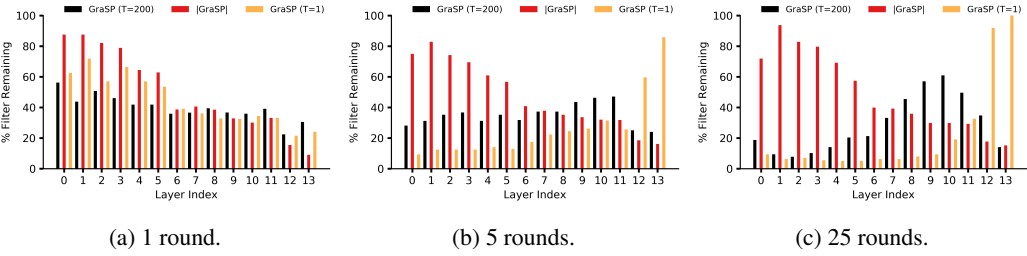

(a) 1 round.     (b) 5 rounds.     (c) 25 rounds.

Figure 17: Layer-wise pruning ratios for MobileNet-V1 with a target ratio of 65% pruning. Results are provided for pruning over (a) 1 round, (b) 5 rounds, and (c) 25 rounds. When the model is allowed to train even slightly, GraSP without temperature prunes earlier layers very aggressively.

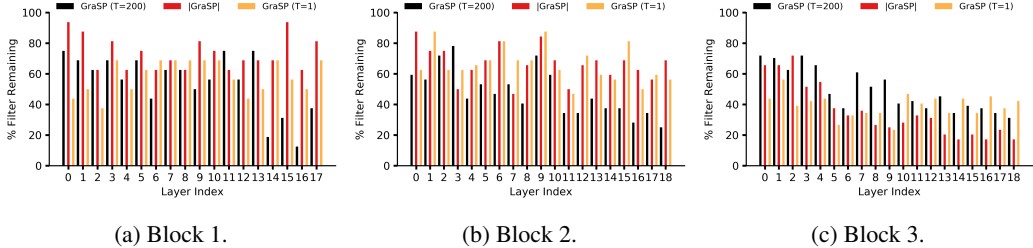

(a) Block 1.  (b) Block 2.  (c) Block 3.

Figure 18: Layer-wise pruning ratios for (a) Block 1, (b) Block 2, and (c) Block 3 of ResNet-56, with a target ratio of 50% pruning over 1 round. For (a) Block 1, even layer indices are Layer-1 of a residual module and odd layer indices are Layer-2 of a residual module; for (b) Block 2 and (c) Block 3, index 0 is Layer-1, index 1 is Layer-2, and index 2 is the learned shortcut layer of the first residual module; thereafter, for (b) Block 2 and (c) Block 3 odd layer indices are Layer-1 of a residual module and even layer indices are Layer-2 of a residual module. This figure shows that when the model is at initialization, GraSP with and without large temperature have only a *slight* bias towards pruning earlier layers more than deeper layers.

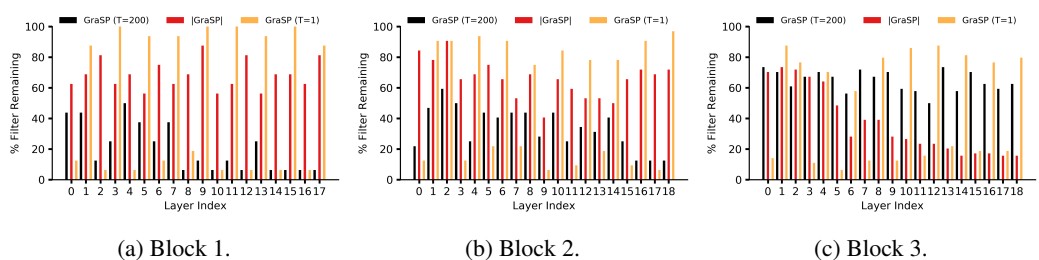

(a) Block 1.  (b) Block 2.  (c) Block 3.

Figure 19: Layer-wise pruning ratios for (a) Block 1, (b) Block 2, and (c) Block 3 of ResNet-56, with a target ratio of 50% pruning over 5 rounds. For (a) Block 1, even layer indices are Layer-1 of a residual module and odd layer indices are Layer-2 of a residual module; for (b) Block 2 and (c) Block 3, index 0 is Layer-1, index 1 is Layer-2, and index 2 is the learned shortcut layer of the first residual module; thereafter, for (b) Block 2 and (c) Block 3 odd layer indices are Layer-1 of a residual module and even layer indices are Layer-2 of a residual module. This figure shows that when the model is allowed to train even slightly, GraSP without large temperature prunes Layer-1's of residual modules in all blocks aggressively, with a clearly higher amount of pruning in the earlier blocks. GraSP with large temperature prunes all earlier layers aggressively.

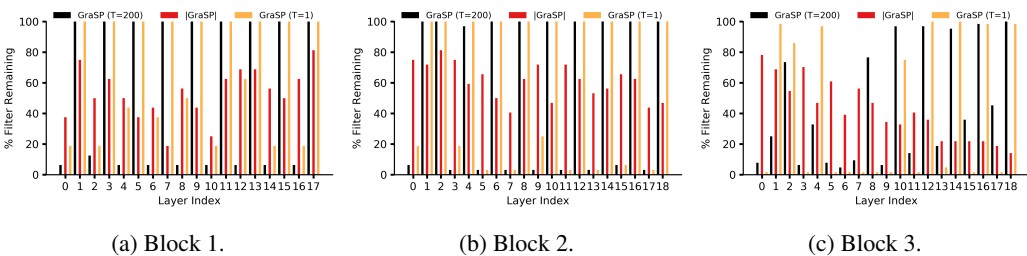

(a) Block 1.  (b) Block 2.  (c) Block 3.

Figure 20: Layer-wise pruning ratios for (a) Block 1, (b) Block 2, and (c) Block 3 of ResNet-56, with a target ratio of 50% pruning over 25 rounds. For (a) Block 1, even layer indices are Layer-1 of a residual module and odd layer indices are Layer-2 of a residual module; for (b) Block 2 and (c) Block 3, index 0 is Layer-1, index 1 is Layer-2, and index 2 is the learned shortcut layer of the first residual module; thereafter, for (b) Block 2 and (c) Block 3 odd layer indices are Layer-1 of a residual module and even layer indices are Layer-2 of a residual module. This figure shows that when the model is allowed to train even slightly, GraSP without temperature prunes Layer-1's of residual modules in all blocks aggressively, with a clearly higher amount of pruning in the earlier blocks. GraSP with large temperature prunes all earlier layers aggressively.

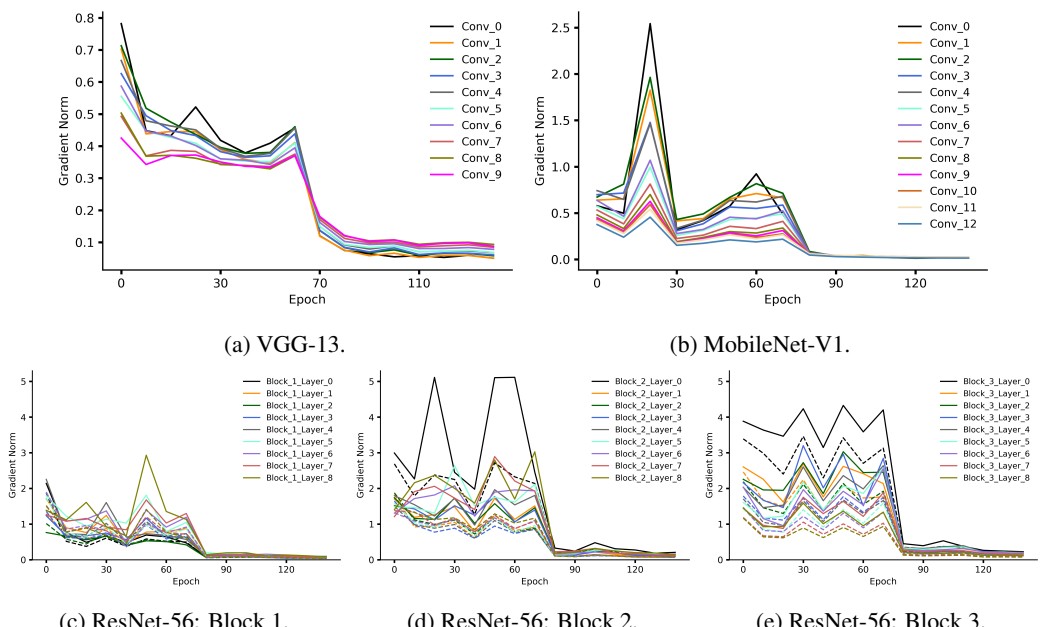

Figure 21: Layer-wise gradient norm for (a) VGG-13, (b) MobileNet-V1, (c) Block 1 of ResNet-56, (d) Block 2 of ResNet-56, and (e) Block 3 of ResNet-56 models trained on CIFAR-100. For MobileNet-V1, number of pointwise convolutional filters in a layer and number of depthwise convolutional filters in the following layer need to be the same. Due to this, we only use pointwise convolutional filters to prune a model and thus this figure visualizes the gradient norm for pointwise convolutional filters only. For ResNet-56 plots, bold lines indicate the first convolutional layer (Layer-1) in a residual module and dotted lines indicate the second convolutional layer (Layer-2) in that same module. Early-on in training, we generally find gradient norm is much higher for earlier layers and these layers are invariably the ones pruned more by GraSP (see layer-wise pruning ratios in Figure 16 for VGG-13 models; Figure 17 for MobileNet-V1 models; and Figure 18, Figure 19, and Figure 20 for ResNet-56 models, respectively).

## G.3 TRAIN/TEST CURVES FOR GRADIENT-NORM BASED PRUNING VARIANTS

This subsection provides train/test curves for different variants of gradient-norm based pruning methods considered in this paper. These curves demonstrate the large temperature GraSP variant does not improve convergence rate substantially, if ever, in comparison to gradient-norm preservation. On the other hand, it does result in significant performance loss when the model is even slightly trained.

### G.3.1 VGG-13

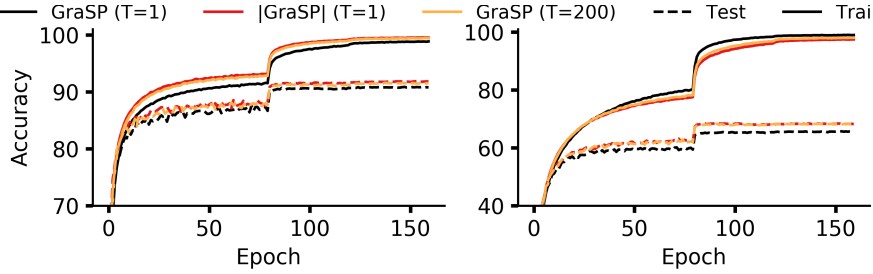

Figure 22: VGG-13: 1 round of pruning. CIFAR-10 (left); CIFAR-100 (right).

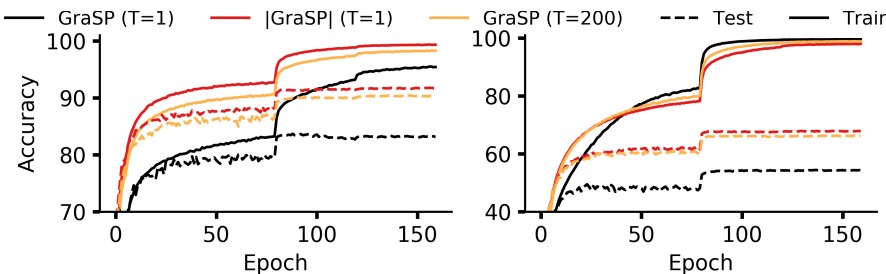

Figure 23: VGG-13: 5 rounds of pruning. CIFAR-10 (left); CIFAR-100 (right).

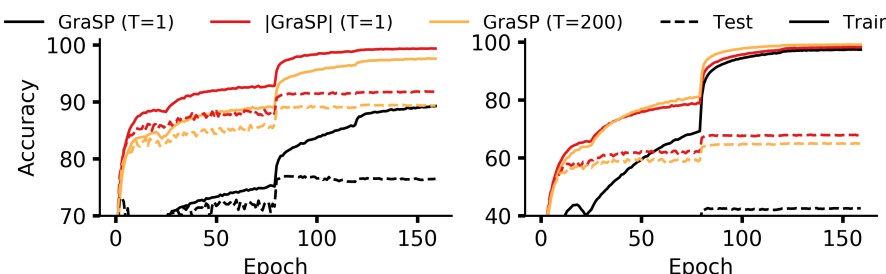

Figure 24: VGG-13: 25 rounds of pruning. CIFAR-10 (left); CIFAR-100 (right).

### G.3.2 MOBILENET-V1

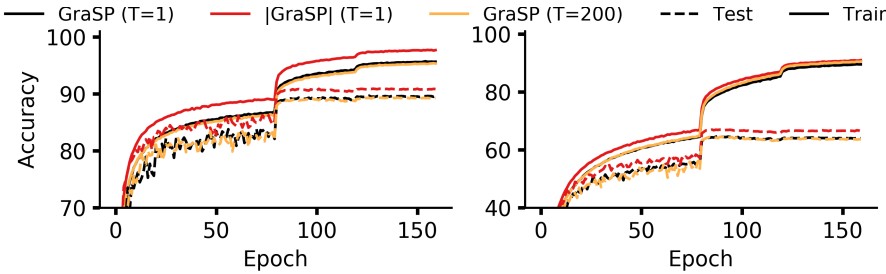

Figure 25: MobileNet-V1: 1 round of pruning. CIFAR-10 (left); CIFAR-100 (right).

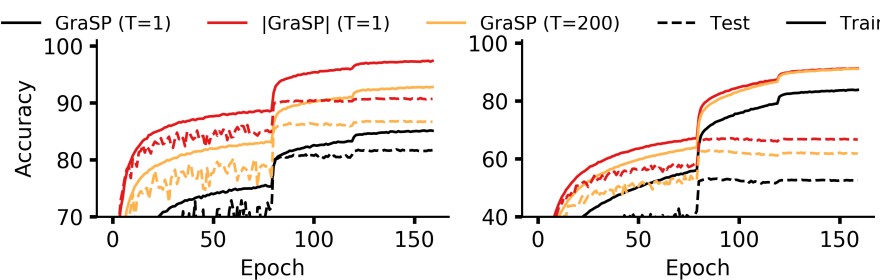

Figure 26: MobileNet-V1: 5 rounds of pruning. CIFAR-10 (left); CIFAR-100 (right).

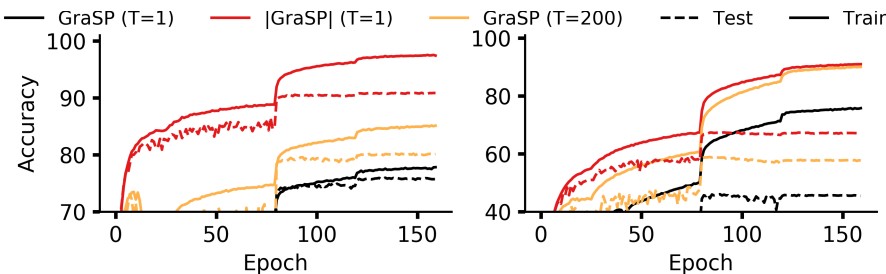

Figure 27: MobileNet-V1: 25 rounds of pruning. CIFAR-10 (left); CIFAR-100 (right).

### G.3.3 RESNET-56

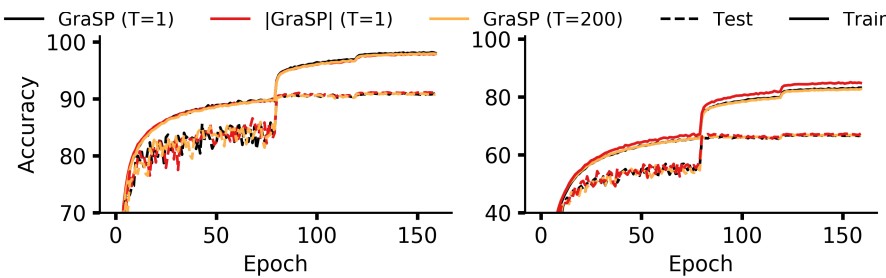

Figure 28: ResNet-56: 1 round of pruning. CIFAR-10 (left); CIFAR-100 (right).

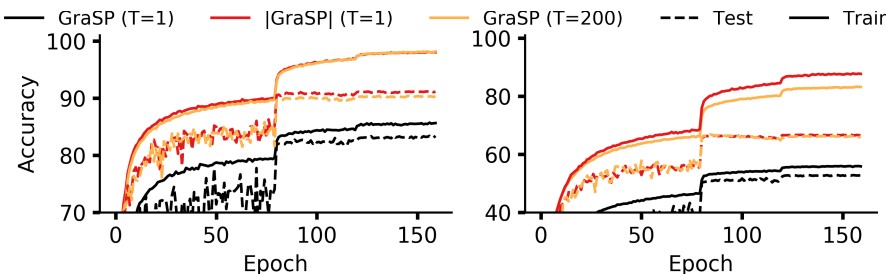

Figure 29: ResNet-56: 5 rounds of pruning. CIFAR-10 (left); CIFAR-100 (right).

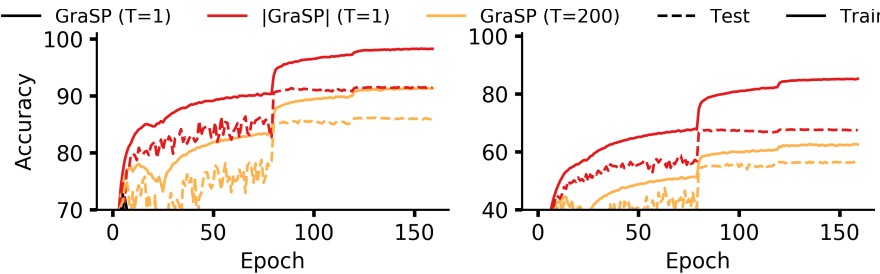

Figure 30: ResNet-56: 25 rounds of pruning. CIFAR-10 (left); CIFAR-100 (right). GraSP (T=1) is not shown because it achieves random performance (10% for CIFAR-10 and 1% for CIFAR-100).

## H    RESULTS ON UNSTRUCTURED PRUNING

The experiments in this paper focus on structured pruning, which enables acceleration on commodity hardware resources without requirement of specially designed hardware/software. However, note that our theoretical analysis is independent of the pruning setup used. Thus, we revisit some of our experiments under an unstructured pruning setup, hence demonstrating the empirical generality of our results.

### H.1    OBSERVATIONS 2/3

In Observation 2 (see Section 4.2), we show that loss-preservation based pruning techniques tend to remove filters with minimal movement in their magnitude. This leads us to Observation 3 (see Section 4.2), which shows that with the added heuristic of "train until minimal change" (You et al. (2020)), where magnitude-based pruning removes parameters with small magnitude and minimal change, magnitude-based pruning implicitly performs loss-preservation as well. The main paper verifies this claim at a filter-level granularity (structured pruning) for models trained on CIFAR-100 (see Figure 2).

In Figure 31, we show that even when analyzed at a finer granularity of parameters (unstructured pruning) in convolutional filters, our claim continues to hold well. In particular, we show that as training continues and change in parameters reduces, $|\boldsymbol{\Theta}_p \Delta \boldsymbol{\Theta}_p|$ becomes highly correlated with loss-preservation based importance ($|\boldsymbol{\Theta}_p(t)\mathbf{g}(\boldsymbol{\Theta}(t))|$). This demonstrates that our observations relating magnitude-based pruning and loss-preservation pruning continue to hold well in an unstructured pruning setup too.

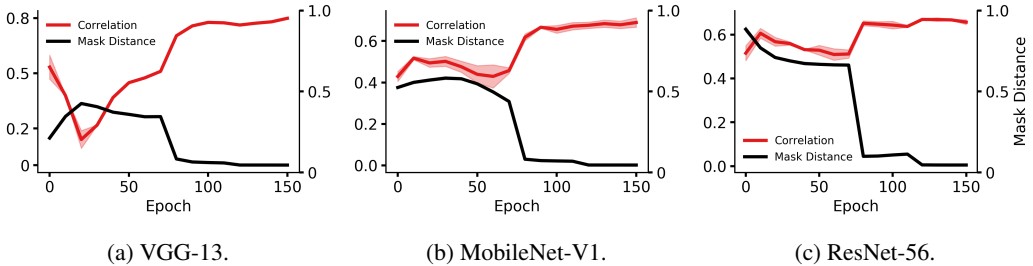

|          (a) VGG-13.          |          (b) MobileNet-V1.          |          (c) ResNet-56.          |

Figure 31: Correlation between $|\boldsymbol{\Theta}_p \Delta \boldsymbol{\Theta}_p|$ and loss-preservation based importance ($|\boldsymbol{\Theta}_p(t)\mathbf{g}(\boldsymbol{\Theta}(t))|$) at every 10th epoch for models trained on CIFAR-100. Also plotted is the distance between pruning masks (target ratio: 20% parameters), similar to masks used by You et al. (2020) to decide when to prune a model. In contrast to You et al., these masks function on a parameter level granularity, as used in unstructured pruning. We note that as the distance between pruning masks over consecutive epochs reduces, $|\boldsymbol{\Theta}_p \Delta \boldsymbol{\Theta}_p|$ becomes more correlated with loss-preservation importance.

### H.2    OBSERVATION 4

Observation 4 indicates gradient-norm based pruning (i.e., GraSP Wang et al. (2020)) removes parameters which maximally increase model loss. The main paper confirms this claim at a filter-level granularity (structured pruning) for CIFAR-100 models (see Figure 3), showing that there exists a high correlation between $\boldsymbol{\Theta}_p(t)\mathbf{H}(\boldsymbol{\Theta}(t))\mathbf{g}(\boldsymbol{\Theta}(t))$ and $\boldsymbol{\Theta}_p(t)\mathbf{g}(\boldsymbol{\Theta}(t))$ for this particular setting.

This subsection generalizes this claim to an unstructured pruning setup by providing scatter plots demonstrating the highly correlated nature of $\boldsymbol{\Theta}_p(t)\mathbf{H}(\boldsymbol{\Theta}(t))\mathbf{g}(\boldsymbol{\Theta}(t))$ and $\boldsymbol{\Theta}_p(t)\mathbf{g}(\boldsymbol{\Theta}(t))$ (see Figure 32 for VGG-13 models, Figure 33 for MobileNet-V1 models, and Figure 34 for ResNet-56 models). In contrast with Section G.1, where plots are visualized on a filter-level granularity, the following plots are visualized on a parameter-level granularity, as expected in an unstructured pruning setup. As can be seen in the plots, the measures are highly correlated throughout model training, indicating gradient-norm increase may severely affect model loss if a partially trained or completely trained model is pruned using $\boldsymbol{\Theta}_p(t)\mathbf{H}(\boldsymbol{\Theta}(t))\mathbf{g}(\boldsymbol{\Theta}(t))$. This demonstrates that our observations for gradient-norm based pruning continue to be valid in an unstructured pruning setup as well.

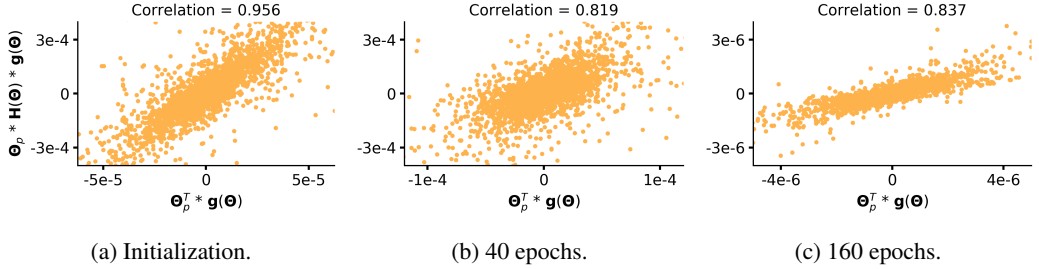

Figure 32: $\mathbf{\Theta}_p(t)\mathbf{H}(\mathbf{\Theta}(t))\mathbf{g}(\mathbf{\Theta}(t))$ versus $\mathbf{\Theta}_p(t)\mathbf{g}(\mathbf{\Theta}(t))$ for VGG-13 models trained on CIFAR-100. Plots are shown for parameters (a) at initialization, (b) after 40 epochs of training, and (c) after complete (160 epochs) training.

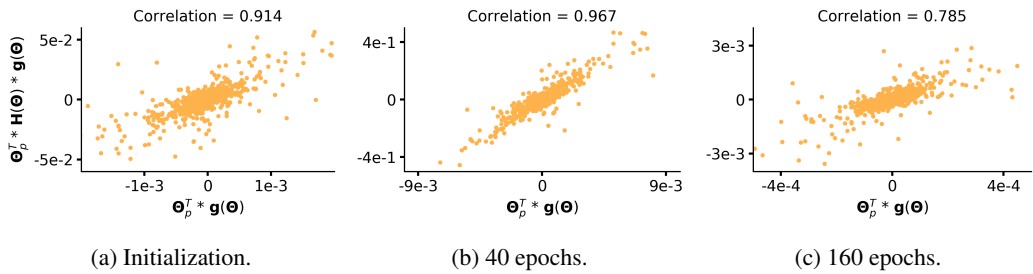

Figure 33: $\mathbf{\Theta}_p(t)\mathbf{H}(\mathbf{\Theta}(t))\mathbf{g}(\mathbf{\Theta}(t))$ versus $\mathbf{\Theta}_p(t)\mathbf{g}(\mathbf{\Theta}(t))$ for MobileNet-V1 models trained on CIFAR-100. Plots are shown for parameters (a) at initialization, (b) after 40 epochs of training, and (c) after complete (160 epochs) training.

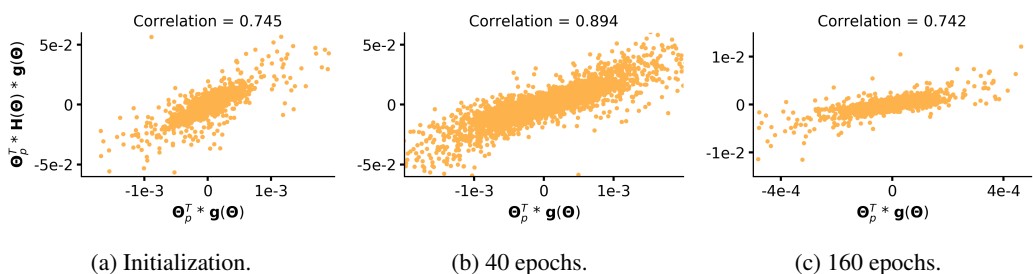

Figure 34: $\mathbf{\Theta}_p(t)\mathbf{H}(\mathbf{\Theta}(t))\mathbf{g}(\mathbf{\Theta}(t))$ versus $\mathbf{\Theta}_p(t)\mathbf{g}(\mathbf{\Theta}(t))$ for ResNet-56 models trained on CIFAR-100. Plots are shown for parameters (a) at initialization, (b) after 40 epochs of training, and (c) after complete (160 epochs) training.

## I    RESULTS ON TINY-IMAGENET

To further verify our claims, we repeat some of our experiments on Tiny-ImageNet and confirm that our claims indeed generalize to the same.

### I.1    OBSERVATIONS 2/3

In Observation 2 (see Section 4.2), we show that loss-preservation based pruning techniques tend to remove filters with minimal movement in their magnitude. This leads us to Observation 3 (see Section 4.2), which shows that with the added heuristic of "train until minimal change" (You et al. (2020)), where magnitude-based pruning removes parameters with small magnitude and minimal change, magnitude-based pruning implicitly performs loss-preservation as well. While the main paper

verifies this claim on CIFAR-100 (see Figure 2), in this section we replicate this behavior on Tiny-ImageNet. As shown in Figure 35, we indeed find that lower the magnitude and movement between BatchNorm scale parameters over subsequent epochs, the higher the correlation between magnitude-based pruning and loss-preservation based pruning is. This demonstrates that our observations relating magnitude-based pruning and loss-preservation pruning generalize to a larger scale dataset too.

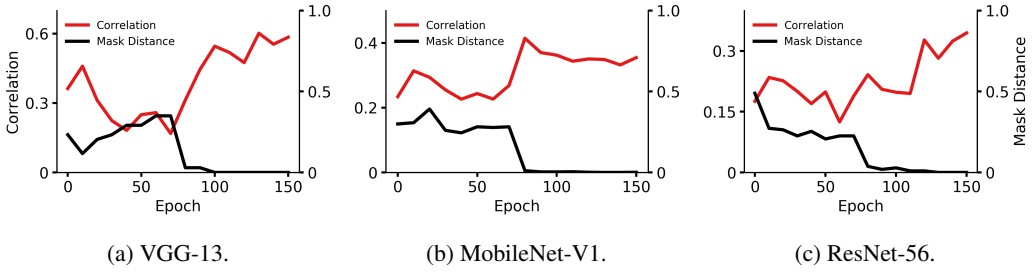

(a) VGG-13.   (b) MobileNet-V1.   (c) ResNet-56.

Figure 35: Correlation between $|\sigma\Delta\sigma|$ (proxy for magnitude-based pruning with the added "train until minimal change" heuristic) and loss-preservation based importance (see Equation 3) at every 10th epoch. Also plotted is the distance between pruning masks (target ratio: 20% filters), as used by You et al. (2020) to decide when to prune a model. As the distance between pruning masks over consecutive epochs reduces, $|\sigma\Delta\sigma|$ becomes more correlated with loss-preservation importance.

### I.2    OBSERVATION 4

Observation 4 indicates gradient-norm based pruning (i.e., GraSP Wang et al. (2020)) removes parameters which maximally increase model loss. The main paper confirms this claim for CIFAR-100 models (see Figure 3), showing that there exists a high correlation between $\mathbf{\Theta}_p^T(t)\mathbf{H}(\mathbf{\Theta}(t))\mathbf{g}(\mathbf{\Theta}(t))$ and $\mathbf{\Theta}_p^T(t)\mathbf{g}(\mathbf{\Theta}(t))$ for this particular setting. Here, we replicate this experiment and confirm our observation on Tiny-ImageNet models (see Figure 36 for VGG-13 models, Figure 37 for MobileNet-V1 models, and Figure 38 for ResNet-56 models).

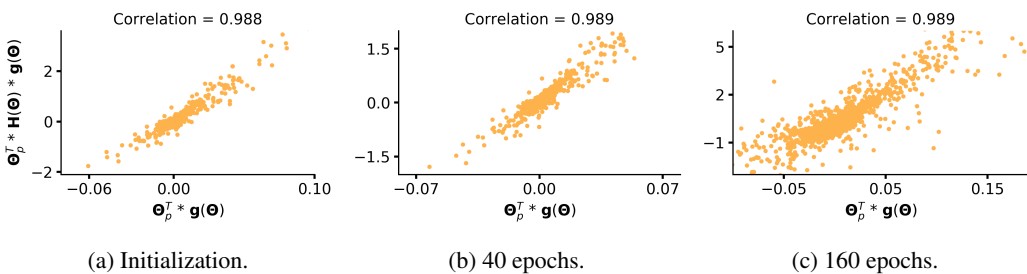

(a) Initialization.   (b) 40 epochs.   (c) 160 epochs.

Figure 36: $\mathbf{\Theta}_p^T(t)\mathbf{H}(\mathbf{\Theta}(t))\mathbf{g}(\mathbf{\Theta}(t))$ versus $\mathbf{\Theta}_p^T(t)\mathbf{g}(\mathbf{\Theta}(t))$ for VGG-13 models trained on Tiny-ImageNet. Plots are shown for filters (a) at initialization, (b) after 40 epochs of training, and (c) after complete (160 epochs) training.

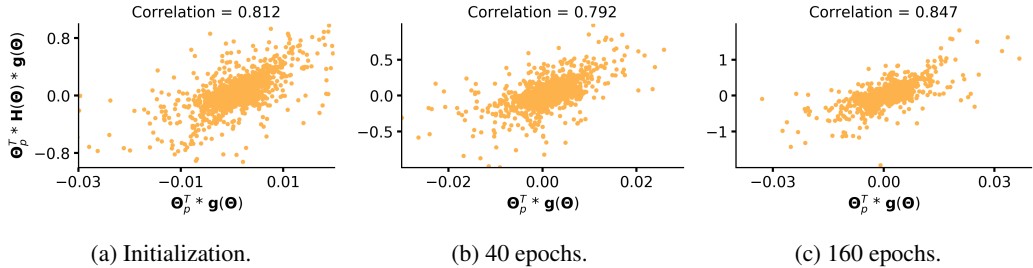

(a) Initialization.

(b) 40 epochs.

(c) 160 epochs.

Figure 37: $\mathbf{\Theta}_p^T(t)\mathbf{H}(\mathbf{\Theta}(t))\mathbf{g}(\mathbf{\Theta}(t))$ versus $\mathbf{\Theta}_p^T(t)\mathbf{g}(\mathbf{\Theta}(t))$ for MobileNet-V1 models trained on Tiny-ImageNet. Plots are shown for filters (a) at initialization, (b) after 40 epochs of training, and (c) after complete (160 epochs) training.

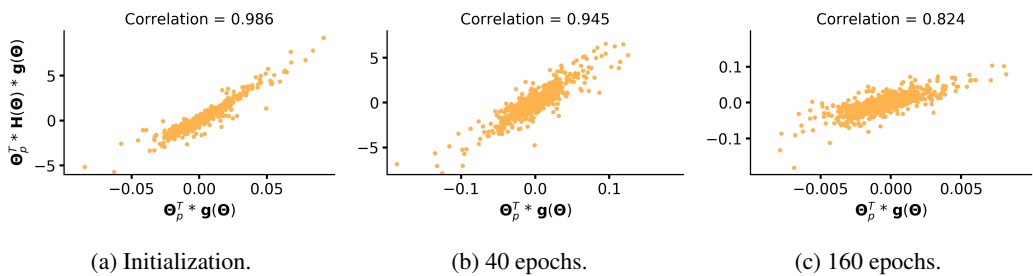

(a) Initialization.

(b) 40 epochs.

(c) 160 epochs.

Figure 38: $\mathbf{\Theta}_p^T(t)\mathbf{H}(\mathbf{\Theta}(t))\mathbf{g}(\mathbf{\Theta}(t))$ versus $\mathbf{\Theta}_p^T(t)\mathbf{g}(\mathbf{\Theta}(t))$ for ResNet-56 models trained on Tiny-ImageNet. Plots are shown for filters (a) at initialization, (b) after 40 epochs of training, and (c) after complete (160 epochs) training.

