# OpenReview forum: "A Gradient Flow Framework For Analyzing Network Pruning"
_ICLR.cc/2021/Conference — ICLR 2021 Spotlight_

### Official Review · AnonReviewer1 · 2020-10-26
**Use of gradient flow to derive interesting relationship and interpretations for pruning early**

**Rating:** 7
**Confidence:** 3

**Review:**

The paper contributes to explaining why saliency measures used for pruning trained models may (or may not) also be effective for pruning untrained or minimally trained models, by developing the relationship between those saliency measures and different forms of the norm of model parameters based on the evolution of model parameters via gradient flow (basically derivatives w.r.t. time). This result leads to several interesting interpretations that could shed some light on on-going efforts to understand recent methods of pruning early-on (e.g., pruning at initialization or after minimal training) and potential extensions to existing saliency measures. The idea of employing gradient flow is novel for its purpose and seems to be accurately executed.

The main concern is that there is a gap between the flow model and the actual optimization method used in this work (SGD with momentum), or more generally standard optimization methods for deep learning. In this regard, the claim of “evolution dynamics” seems a bit exaggerated and remains as theoretical; experiments are strictly speaking not entirely valid to support it either. (minor) Related work is written as if pruning is only done via saliency-based methods (e.g., “pruning frameworks generally define importance measures”) without taking into account various others such as optimization based methods employing sparsity inducing penalty terms. On a different but related note, the writing becomes a bit loose when it comes to referencing existing methods; it is worth correcting them and clarifying the scope/focus of this work.

Further questions:
- Why do you study structured pruning *only*? The provided reasons (“unstructured pruning requires specially designed hardwares or softwares” or “higher relevance to practitioners”) don’t seem valid enough if the purpose really lies in analyzing. Can you provide any results for unstructured pruning?
- Can you provide evidence to support the claim “GraSP without large temperature chooses to prune earlier layers aggressively” (besides Raghu et al. 2017)?
- Based on Tables 1 and 2 the proposed extension to loss-preservation method works the best, while the differences across different methods seem a bit marginal. Is my understanding correct?

---

> ### Author Response · Authors · 2020-11-18
> **Thank you for your valuable feedback! Responses to specific comments are provided below.**
>
> Thank you for your valuable feedback! Responses to specific comments are provided below.
>
> **Response to Main concerns:**
>
> 1. *The main concern is that there is a gap between the flow model and the actual optimization method used in this work (SGD with momentum), or more generally standard optimization methods for deep learning. In this regard, the claim of "evolution dynamics" seems a bit exaggerated and remains as theoretical; experiments are strictly speaking not entirely valid to support it either.*
>
> To address this point, we've added a theoretical analysis of evolution of norm of model parameters under the commonly used SGD training method (see Appendix B). Therein, we show the expected first- and second-order time derivative of norm of model parameters for SGD training exactly match the first- and second-order time derivatives of norm of model parameters for gradient flow. The involved assumption is that terms beyond $\mathcal{O}(\eta^{2})$ and $\mathcal{O}(\eta^{3})$, where $\eta$ is the learning rate, can be ignored, as is generally assumed in development of the importance measures analyzed in this work. This implies our claims are not just theoretical, but in fact provide a valid approximation to SGD training.
>
> 2. *(minor) Related work is written as if pruning is only done via saliency-based methods (e.g., "pruning frameworks generally define importance measures") without taking into account various others such as optimization based methods employing sparsity inducing penalty terms. On a different but related note, the writing becomes a bit loose when it comes to referencing existing methods; it is worth correcting them and clarifying the scope/focus of this work.*
>
> Thank you for pointing this out! We've updated the related work section to indicate our work focuses on pruning frameworks that specifically involve the use of an importance measure.
>
> **Response to Further questions:**
>
> 1. *Why do you study structured pruning only? The provided reasons ("unstructured pruning requires specially designed hardwares or softwares" or "higher relevance to practitioners") don't seem valid enough if the purpose really lies in analyzing. Can you provide any results for unstructured pruning?*
>
> Structured pruning faces fewer barriers to widespread use. By clarifying benefits and pitfalls of existing importance measures, we aim at enabling practitioners with limited resources to make informed choices on how to minimize their network training expenses. Most works on pruning early-on in training also share this motivation, but ultimately provide results under unstructured pruning setups, which do not translate to execution efficiency directly.
>
> However, we agree that testing our claims in the context of unstructured pruning is valuable. Appendix H now contains experimental results demonstrating validity of our importance measure claims (Observations 2, 3, and 4) at parameter-level granularity, i.e., for unstructured pruning. We also refer the reviewer to a parallel submission at ICLR (https://openreview.net/forum?id=Ig-VyQc-MLK), which independently arrives at one of our claims (Observation 5) through an empirical analysis and shows its validity in unstructured pruning settings.
>
> 2. *Can you provide evidence to support the claim "GraSP without large temperature chooses to prune earlier layers aggressively" (besides Raghu et al. 2017)?*
>
> We've added layer-wise pruning ratios for different models in Appendix G.2 (see Figures 16–20). The plots clearly indicate that in comparison to other variants, GraSP without temperature prunes earlier layers more aggressively.
>
> In the same section (Appendix G.2), we also use gradient flow to show that since reduction in model loss is proportional to gradient norm, the layer-wise gradient norm (see Figure 21) can be a good indicator of which layers are most responsible for reduction in model loss. Early-on in training, gradient-norm is generally highest for earlier layers. This indicates reduction in model loss early-on in training can indeed be attributed to earlier layers, as also claimed by Raghu et al., 2017 through the use of SVCCA.
>
> 3. *Based on Tables 1 and 2 the proposed extension to loss-preservation method works the best, while the differences across different methods seem a bit marginal. Is my understanding correct?*
>
> Generally, we found most metrics perform equally well, while both magnitude-based pruning and the extended measure work the best. On several experiments, magnitude-based pruning outperforms the proposed extension and vice versa.

---

### Official Review · AnonReviewer3 · 2020-10-27
**Excellent, clear paper with compelling insights and empirical results**

**Rating:** 9
**Confidence:** 5

**Review:**

Summary:

The authors study proposed importance metrics for pruning neurons/channels in deep neural networks and analyze what properties of parameters are favored by each approach by studying the relationship between model parameters, gradients, 2nd order derivatives and loss. Through this analysis they develop a rich understanding of the consequences of different pruning criteria and use their understanding to propose modifications to existing techniques that produce higher quality models across different settings.

Pros:

The framework used by the authors is clear and easy to understand but also very general. The authors’ mix of empirical results and theoretical analysis makes a convincing case for the accuracy of their observations. The authors go beyond observation and analysis and use their insights to design new approaches to pruning that outperform existing techniques. The paper is well written and well organized.

Cons:

This paper has few limitations. The main limitation is that all experiments were conducted on relatively small datasets (CIFAR). Given that is has been shown that some techniques in model compression produce state-of-the-art results on small tasks but fail on larger models and datasets [1, 2], I’d encourage their authors to further validate their insights on a larger dataset (i.e., ImageNet).

Comments:

I found that the authors waited a long time to explain the term “gradient flow”, which was important in sections 1-3 but not fully detailed until the start of section 4. On page 1 the authors say in parenthesis that gradient flow is “gradient descent with infinitesimal learning rate”, but I found this explanation was not clear. The second sentence of section 4 “the evolution over time of model parameters, gradient, and loss” was much more clear. I’d encourage the authors to potentially work some of these details earlier into the text.

References:
1. https://arxiv.org/abs/1902.09574
2. https://arxiv.org/abs/2003.03033

---

> ### Author Response · Authors · 2020-11-18
> **Thank you for your valuable feedback! Responses to specific comments are provided below.**
>
> Thank you for your valuable feedback! Responses to specific comments are provided below.
>
> 1. *This paper has few limitations. The main limitation is that all experiments were conducted on relatively small datasets (CIFAR). Given that is has been shown that some techniques in model compression produce state-of-the-art results on small tasks but fail on larger models and datasets [1, 2], I'd encourage their authors to further validate their insights on a larger dataset (i.e., ImageNet).*
>
> Please see Appendix I. We have added further verifications on Tiny-ImageNet for our observations related to behaviors of different importance measures (Observations 2, 3, and 4). For Observation 5, we refer the reviewer to a parallel submission at ICLR (https://openreview.net/forum?id=Ig-VyQc-MLK) which independently arrives at Observation 5 through empirical analysis and verifies it on ImageNet in an unstructured pruning setting.
>
> 2. *I found that the authors waited a long time to explain the term "gradient flow" which was important in sections 1--3 but not fully detailed until the start of section 4.*
>
> Thank you for pointing this out! We have added a footnote and a forward reference to another section that provides a short primer on gradient flow.

---

> > ### Comment · AnonReviewer3 · 2020-11-24
> > **Reviewer Response 1**
> >
> > Thank you to the authors for their response. I think the paper should be accepted, and the feedback below is just general feedback to help improve the paper further.
> >
> > The additional experiments are helpful. A non-vision task like WikiText-103 with a Transformer model would also be very helpful to show that the observation generalize across model classes. I would still encourage the authors to pursue larger datasets e.g., full ImageNet for their existing experiments.
> >
> > A footnote and a forward reference is helpful, but I'd recommend a re-structuring of the text to make the paper more clear. It is a flaw in the writing if the reader has to jump around to understand what's going on.

---

### Official Review · AnonReviewer4 · 2020-10-28

**Rating:** 6
**Confidence:** 5

**Review:**

## Summary
This paper studies different families of pruning criteria and their impact on training dynamics (especially early training). Stemming from the observations, authors provide improvements to the 1st and 2nd order saliency methods.

## Pros
- Authors provide simple and useful explanations to various pruning criteria that are based on the Taylor approximation of the loss functions.
- Even the authors don't mention this in the contributions, they propose some improved versions of existing criteria. For example the updated taylor score with $\theta^2g(\theta)$ or absolute valued GrasP. This is great and it might worth focusing on these criteria further providing further evidence on their usefulness. Currently, they seem a bit arbitrary. For example, why not third power $\theta^3g(\theta)$ or additive biasing of magnitude $(g(\theta)+c)*\theta$. I recommend authors to run their versions in unstructured setting too.

## Cons
- Authors choose to focus on structured pruning since resulting networks are dense and acceleration is straight-forward. However,  they miss an important work on structured pruning [1]. This relatively well-known work shows that pruned (structured) networks can be trained to full accuracy from scratch. In other words, their value lies on doing some kind of an architecture search over layer widths. The motivation of the work needs to be revisited in the light of these results. Since we can retrain pruned networks from scratch, it probably doesn't matter which neuron we choose and therefore which criteria is better. Unstructured pruning doesn't have this training from scratch issue, and I recommend authors to at least include and maybe shift the focus to unstructured pruning.
- "but requires specially designed hardware (Han et al. (2016a)) or software (Elsen et al. (2020)). While results in this paper are applicable in both settings, our experimental evaluation focuses on structured pruning due to its higher relevance to practitioners." All neural networks require special hardware if you want to accelerate them. I think a better motivation here is to point out to the difficulties at accelerating sparse operations and limited availability/support for such operations in existing frameworks. And I am not sure how useful structured pruning algorithms are given the results of [1].
- "The larger the magnitude of parameters at a particular instant, the smaller the model loss at that instant will be." This is likely to be true in simple settings, however it is not a sufficient condition; especially for networks with batch norm. You can arbitrarily scale neurons if there is a batch-norm and you can come-up with arbitrary ordering if needed. I recommend re-phrasing this observation and/or stating the assumptions better (I don't remember seeing any assumption on the network itself). How the regularization or gradient noise will effect this statement?
- "Thus, the parameter with the most negative value for Θ(t)g(Θ(t)) is likely to also have a large, negative value for Θ(t)H(Θ(t))g(Θ(t))" This is not clear to me. Assume 1d case where Θ(t)= -1; g(Θ(t))=2; H(Θ(t))=-1 -> Θ(t)g(Θ(t))=-2; Θ(t)H(Θ(t))g(Θ(t))=2. I can see the correlation in the figure but it doesn't seem like an obvious thing. Maybe because hessian don't have many negative eigenvalues?

## Rating
I found the results and analysis interesting, however motivation needs to be updated. The work would also benefit from including unstructured pruning experiments.

## Minor Points
- "Recent works focus on pruning models at initialization (Frankle & Carbin (2019);..." Lottery Ticket paper prunes after training and show existence of some initializations that achieve good performance..
-  Equations 6/7 $\frac{dL}{dt}= ||g(\theta)||^2$ assuming gradient descent shouldn't be a learning rate?
- "...than magnitude-agnostic techniques." Which methods are these? As far as I see, all methods use magnitude information in their formulas directly or indirectly.
- In Table:1, I recommend authors to bold both scores if they lie within the std of each other; so that we can identify which improvements are significant.
- It would be nice to show how the temperature parameter is used for GrasP

[1] https://arxiv.org/abs/1810.05270

---

> ### Author Response · Authors · 2020-11-18
> **Thank you for your valuable feedback! Responses to specific comments are provided below.**
>
> Thank you for your valuable feedback! Responses to specific comments are provided below.
>
> **Response to Cons**
>
> 1. *Authors choose to focus on structured pruning since resulting networks are dense and acceleration is straight-forward. However, they miss an important work on structured pruning [1]. This relatively well-known work shows that pruned (structured) networks can be trained to full accuracy from scratch. In other words, their value lies on doing some kind of an architecture search over layer widths. The motivation of the work needs to be revisited in the light of these results. Since we can retrain pruned networks from scratch, it probably doesn't matter which neuron we choose and therefore which criteria is better.*
>
> Thank you for bringing [1] to our attention. We were aware of that work, but do not believe that it weakens our work's motivation. That paper claims that structured pruning frameworks help design architectures that are capable of achieving high accuracy, but preserving the weights of a pruned model after pruning is unnecessary. However, the authors never imply that the choice of importance measure used for structured pruning doesn't matter. In fact, as per your claim ("it probably doesn't matter which neuron we choose and therefore which criteria is better"), if it indeed does not matter which criteria are used for structured pruning, even random or uniform pruning should work well. To demonstrate that this is not the case, we have added experimental results on random/uniform pruning to Appendix E. These results demonstrate that both uniform and random pruning perform much worse (4—5% lower accuracy, frequently) than a standard pruning measure based on magnitude, loss preservation, or gradient norm.
>
> Nonetheless, we agree that evaluating our claims in the context of unstructured pruning is also valuable. Thus, we've added experiments (see Appendix H) demonstrating validity of our claims related to importance measures (Observations 2, 3, and 4) at parameter-level granularity, i.e., for unstructured pruning. We also refer the reviewer to a parallel submission at ICLR (https://openreview.net/forum?id=Ig-VyQc-MLK), which independently arrives at one of our claims (Observation 5) through empirical analysis and shows its validity in the context of unstructured pruning.
>
> 2. *"but requires specially designed hardware (Han et al. (2016a)) or software (Elsen et al. (2020)). While results in this paper are applicable in both settings, our experimental evaluation focuses on structured pruning due to its higher relevance to practitioners." All neural networks require special hardware if you want to accelerate them. I think a better motivation here is to point out to the difficulties at accelerating sparse operations and limited availability/support for such operations in existing frameworks.*
>
> Thanks! We have changed the phrasing of that paragraph to more precisely express our intent (found in Section 2: Related Work): "From an implementation standpoint, pruning approaches can be placed in two categories. The first, structured pruning ((Li et al. (2017); He et al. (2018); Liu et al. (2017); Molchanov et al. (2019; 2017); Gao et al. (2019))), removes entire filters, thus preserving structural regularity and directly improving execution efficiency on commodity hardware platforms. The second, unstructured pruning (Han et al. (2016b); LeCun et al. (1990); Hassibi \& Stork (1993)), is more fine-grained, operating at the level of individual parameters instead of filters. Unstructured pruning has recently been used to reduce computational complexity as well, but requires specially designed hardware (Han et al. (2016a)) or software (Elsen et al. (2020)) that are capable of accelerating sparse operations. By clarifying benefits and pitfalls of existing importance measures, our work aims to ensure practitioners are better able to make informed choices for reducing DNN training/inference expenditure via network pruning. Thus, while results in this paper are applicable in both structured and unstructured settings, our experimental evaluation primarily focuses on structured pruning early-on in training."
>
> (Continued in next comment.)

---

> > ### Author Response · Authors · 2020-11-18
> > **Response (part 2)**
> >
> > 3. *"The larger the magnitude of parameters at a particular instant, the smaller the model loss at that instant will be." This is likely to be true in simple settings, however it is not a sufficient condition; especially for networks with batch norm. You can arbitrarily scale neurons if there is a batch-norm and you can come-up with arbitrary ordering if needed. I recommend re-phrasing this observation and/or stating the assumptions better (I don't remember seeing any assumption on the network itself). How the regularization or gradient noise will effect this statement?*
> >
> > Please refer to Du et al., 2018 (https://arxiv.org/pdf/1806.00900.pdf). That paper demonstrates deep homogeneous models (models with ReLU activations) trained using gradient flow are constrained by an implicit regularization mechanism which balances the norms of model parameters across layers. That is, the model parameters evolve such that difference of norm of layer-wise parameters remains constant. A more recent paper provides a variant of that result for SGD-trained models (see Figure 5 and Equation 18 of https://openreview.net/forum?id=q8qLAbQBupm), demonstrating that the layer-wise norms of parameters obey the same balanced-norm property with an additional term that exponentially decays over time.
> >
> > Therefore, when a model is allowed to train naturally under gradient descent variants (such as SGD or gradient flow), an unconstrained update that uniformly scales all parameters of a layer to increase their magnitude is prevented by the implicit regularization mechanism. Thus, to respond to your specific comment on what conditions enable our observation to hold well, we simply require a model to be optimized through a standard gradient descent framework. As clarified in the beginning of Section 4, we study model evolution under gradient flow and this is a constant assumption throughout our paper. An artificial perturbation, such as the suggested scaling up of all parameters, does not occur naturally in this regime.
> >
> > 4. *"Thus, the parameter with the most negative value for Θ(t)g(Θ(t)) is likely to also have a large, negative value for Θ(t)H(Θ(t))g(Θ(t))" This is not clear to me. Assume 1d case where Θ(t)= -1; g(Θ(t))=2; H(Θ(t))=-1 -> Θ(t)g(Θ(t))=-2; Θ(t)H(Θ(t))g(Θ(t))=2. I can see the correlation in the figure but it doesn't seem like an obvious thing. Maybe because hessian don't have many negative eigenvalues?"*
> >
> > We agree about your observation, but note that we used the term "likely", not "necessarily", in the quoted comment. Rarely, but some times, a quantity changes with negative acceleration and positive velocity. Such cases exist in our reported experiments as well. For example, in Figure 3, where scatter plots of Θ(t) H(Θ(t)) g(Θ(t)) versus Θ(t) g(Θ(t)) are shown, one can see the fourth quadrant of the plots has a few points. These points have a positive value for Θ(t) g(Θ(t)), but negative value for Θ(t) H(Θ(t)) g(Θ(t)). However, our claim implies that if a quantity has negative acceleration, it is likely to have negative velocity too. For majority of the model parameters, this holds well: negative Θ(t) H(Θ(t)) g(Θ(t)) and negative Θ(t) g(Θ(t)) occur simultaneously for most parameters. In fact, the correlation between the two metrics reaches 0.95—0.99 in several cases.
> >
> > (Continued in next comment.)

---

> > > ### Author Response · Authors · 2020-11-18
> > > **Response (part 3)**
> > >
> > > **Response to Minor Points**
> > >
> > > 1. *"Recent works focus on pruning models at initialization (Frankle \& Carbin (2019);..." Lottery Ticket paper prunes after training and show existence of some initializations that achieve good performance..*
> > >
> > > Thank you for pointing this out! We've rephrased this statement.
> > >
> > > 2. "Equations 6/7 $\frac{dL}{dt} = -||g(\theta)||^{2}$ assuming gradient descent shouldn't be a learning rate?"
> > >
> > > Note that gradient flow is the infinitesimal variant of gradient descent, and therefore $dt$ is the learning rate. To be more precise, if a model has parameters $\theta$, then loss evolution under gradient descent up to a first-order Taylor approximation is:
> > >
> > > $L(\theta - \eta g(\theta)) - L(\theta) = -\eta g(\theta)^{T} g(\theta) = - \eta ||g(\theta)||^{2}$.
> > >
> > > Define $dt := \lim_{\eta \rightarrow 0}$, then we have:
> > >
> > > $dL = L(\theta - dt g(\theta)) - L(\theta) = - dt ||g(\theta)||^{2} \implies \frac{dL}{dt} = -||g(\theta)||^{2}$.
> > >
> > > 3. *"...than magnitude-agnostic techniques." Which methods are these? As far as I see, all methods use magnitude information in their formulas directly or indirectly.*
> > >
> > > This seems to be a disagreement about definitions. While the derivative of a function may be dependent on the function value, its mathematical definition isn't directly dependent on the function's value. Our work shows the importance measure used for loss-preservation based pruning is equal to the first-order time-derivative of norm of model parameters, not the norm itself. Thus, we call such a method magnitude-agnostic—i.e., they are dependent on properties (e.g., first-order time derivative) related to evolution of the magnitude, but not the magnitude itself.
> > >
> > > We have also added a brief footnote under Observation 2 of our paper to make this point clear: "This observation also implies loss preservation’s importance measure does not depend on magnitude of parameters directly. We thus call loss-preservation a magnitude-agnostic technique."
> > >
> > > 4. *It would be nice to show how the temperature parameter is used for GrasP*
> > >
> > > Thank you for this comment! We've added further details and a reference to Appendix G: "For the original GraSP variant that increases gradient norm, we follow the original implementation and use a temperature of $200$ during the calculation of Hessian-gradient product. This involves division of the model output (i.e., the logits vector) by a constant T ($=200$) before using softmax for classification (see Hinton et al. (2015) for further details)."

---

> ### Comment · AnonReviewer4 · 2020-11-23
> **Thank you for rebuttal / updated score & remaining concern.**
>
> I like to thank authors for taking time on the rebuttal and adding (at least) 3 appendix section that addresses various questions/concerns I had. Most of my concerns are addressed (therefore raised score), however, a high-level concern remains about the value of structured pruning;
>
> - **structured pruning**; I still don't see any reference to [1] nor a discussion about it. Experiments in Appendix-E is seems insufficient/expected and they are not the baselines proposed in [1]. Of course pruning criteria matters when you prune a trained network. But do you need to prune a trained network? Instead can you just train the small architecture found ( possibly non-uniform shrinking)? [1] shows that the answer is 'yes' for the major methods of the time and shows the importance of having simple baselines when evaluating pruning criteria. This baselines are called Scratch-E and Scratch-B and I think having these baselines along with a discussion would add great value to the usefulness of the pruning criteria.
> [2] https://arxiv.org/abs/1911.11134

---

> > ### Author Response · Authors · 2020-11-24
> > **Thank you for your response! Updated discussion in related work.**
> >
> > Thank you for your response!
> >
> > Your original review contains comments such as "Since we can retrain pruned networks from scratch, it probably doesn't matter which neuron we choose and therefore which criteria is better", which we believe implies even random pruning on minimally (untrained/partially trained) trained models should result in architectures capable of performing as well as a more sophisticated criteria (e.g., loss preservation based pruning). This indeed isn't the case, which we show in Appendix E. Based on your response, we believe this is your position as well, that different pruning criteria indeed matter. Thus, we may have misinterpreted your original comment related to a discussion on [1]. Our current interpretation of your comments is that since authors in [1] show reinitializing pruned models, as extracted from trained networks, does not result in performance loss, the value of network pruning lies in determining an optimal architecture that can support both high accuracy and high computational efficiency. By discussing [1] along with our already reported demonstration of why pruning criteria designed for trained models are useful on untrained/partially trained models as well, the motivation of our work can be presented more strongly. There's a chance we may still be misinterpreting your comments and, if so, we would be grateful for further clarification.
> >
> > Currently, we've updated the second paragraph of Section 2 (Related work) as follows to reflect our interpretation of your comments and to discuss [1]:
> > *Despite its success, the foundations of network pruning are not well understood. Recent work has shown that good "subnetworks" that achieve similar performance to the original network exist within both trained  (Ye et al. (2020)) and untrained models (Frankle & Carbin (2019); Malach et al. (2020)). These works thus prove networks can be pruned without loss in performance, but do not indicate how a network should be pruned, i.e, which importance measures are preferable. In fact, Liu et al. (2019) show reinitializing pruned models before retraining rarely affects their performance, indicating the consequential differences among importance measures are in the properties of architectures they produce. Since different importance measures perform differently (see Appendix E), analyzing popular measures to determine which model properties they tend to preserve reveal which measures can result in better-performing architectures.*

---

### Official Review · AnonReviewer2 · 2020-11-01
**novelty & experiment at scale**

**Rating:** 6
**Confidence:** 4

**Review:**

This paper proposes a detailed analysis on pruning heuristics, and its applications to early pruning. It thoroughly analyzed magnitude-based pruning, loss-preservation based pruning, and gradient-norm based pruning. The paper demonstrated the results on CIFAR-10 and CIFAR-100 datasets. it's very timely research to guide the audience which heuristic is better. My major concern is the novelty over existing pruning heuristics, since the techniques have all been proposed before. The other concern is the evaluation and the scale of the dataset. Given the results in table 2 different by less than a percent, and Cifar training is very noisy, it's hard to tell the difference. Just like the Lottery Ticket hypothesis works on Cifar but does not work on ImageNet, different pruning heuristics needs to be verified on the large scale ImageNet dataset in order to be convincing.

---

> ### Author Response · Authors · 2020-11-18
> **Thank you for your valuable feedback! Responses to specific comments are provided below.**
>
> Thank you for your valuable feedback! Responses to specific comments are provided below.
>
> 1. *My major concern is the novelty over existing pruning heuristics, since the techniques have all been proposed before.*
>
> Our main objective in this work is not to propose new pruning heuristics, but to analyze the benefits and limitations of existing pruning heuristics in a theoretically grounded manner—e.g., during early phases of this work, we noticed that GraSP is pathologically aggressive in its pruning of early layers if the model is even slightly trained. Through our analysis, we are able to provide a grounded explanation for this behavior. Such expositions, if published, will enable practitioners to avoid inherent weaknesses in existing pruning techniques. In fact, our work's contributions can be very well summarized by quoting your own comment, "This paper proposes a detailed analysis on pruning heuristics, and its applications to early pruning. $\dots$ It's very timely research to guide the audience which heuristic is better."
>
> 2. *The other concern is the evaluation and the scale of the dataset. Given the results in table 2 different by less than a percent, and Cifar training is very noisy, it's hard to tell the difference.*
>
> Our experimental results show that substantial differences are common.  Table 2 shows results for different variants of GraSP: (a) our proposed variant; (b) GraSP with large temperature (as originally proposed by the work's authors); and (c) GraSP without temperature. The table lists 18 experiments. One-third of these have a $>$4.5\% accuracy gap between our proposed variant of GraSP and the next best performing method. Since the margin of difference is large on many settings, we disagree that our results can be attributed to noisy training on datasets only.
>
> 3. *Just like the Lottery Ticket hypothesis works on Cifar but does not work on ImageNet, different pruning heuristics needs to be verified on the large scale ImageNet dataset in order to be convincing.*
>
> Please see Appendix I. We have added further verifications on Tiny-ImageNet for our observations related to behaviors of different importance measures (Observations 2, 3, and 4). For Observation 5, we refer the reviewer to a parallel submission at ICLR (https://openreview.net/forum?id=Ig-VyQc-MLK) which independently arrives at Observation 5 through an empirical analysis and verifies it on ImageNet in unstructured pruning settings.

---

### Decision · Program_Chairs · 2021-01-07
**Final Decision**

**Decision:**

Accept (Spotlight)

**Comment:**

This paper proposes a broad framework for unifying various pruning approaches and performs detailed analyses to make recommendations about the settings in which various approaches may be most useful. Reviewers were generally excited by the framework and analyses, but had some concerns regarding scale and the paper's focus on structured pruning. The authors included new experiments however, which mostly addressed reviewer concerns. Overall, I think is a strong paper which will likely be provide needed grounding for pruning frameworks and recommend acceptance.